# RIESZ NEURAL OPERATOR FOR SOLVING PARTIAL DIFFERENTIAL EQUATIONS

**Shouyi Liu**[1,2]**, Xiaokang Yang**[1]**, Yuntian Chen**[2]*

[1]Shanghai Jiao Tong University      [2]Eastern Institute of Technology

`shouyiliu@sjtu.edu.cn, xkyang@sjtu.edu.cn, ychen@eias.ac.cn`

## ABSTRACT

Local non-stationarity is pivotal to solving partial differential equations (PDEs). However, in operator learning, the spatially local information inherent in the data is often overlooked. Even when explicitly modeled, it is usually collapsed into local superpositions within the model architecture, preventing full exploitation of local features in physical phenomena. To address this limitation, our paper proposes a novel Riesz Neural Operator (RNO) based on the spectral derivative representation. Since PDEs are fundamentally governed by local derivatives, RNO leverages the Riesz transform, a natural spectral representation of derivatives, to mix global spectral information with local directional variations. This approach allows the RNO to outperform existing operators in complex scenarios that require sensitivity to local detail. Our design bridges the gap between physical interpretability and local dynamics. Experimental results demonstrate that the RNO consistently achieves superior prediction accuracy and generalization performance compared to existing approaches across various benchmark PDE problems and complex real-world datasets, presenting superior non-linear reconstruction capability in model analysis.

## 1 INTRODUCTION

Accurately simulating physical systems governed by partial differential equations (PDEs) remains foundational across fluid dynamics (Herde et al., 2024), materials science (Li et al., 2024), and climate modeling (Kurth et al., 2023). Neural operators have recently emerged as powerful surrogates for PDE solution operators, offering significant reductions in compute time and broad adaptability. However, it is a complex task to capture and predict the non-linear and local changes due to physical information missing, transform without local emphasis, etc. This work leverages local spectral derivatives to improve the precision of PDE solutions.

Existing operator-learning frameworks, including DeepONet (Lu et al., 2021; Gu et al., 2025), the Fourier Neural Operator (FNO) (Li et al., 2020a), and the Laplace Neural Operator (LNO) (Cao et al., 2024a), leverage specific transforms to model PDE dynamics. Suppose we analogize the process of function learning to the performance of a musical score. In DeepONet, each partitioned ensemble (trunk function) is assigned a fixed section, which plays as required by the score. This design elegantly captures the global melody of function-to-function mappings. However, it lacks control over melodic variations and requires an excessively large ensemble for complex scores. FNO, by contrast, assigns finer-grained roles with varying intensities to all performers but struggles to capture melodic diversity and local harmonies. LNO augments FNO with a smoother, yielding greater stability at the cost of detailed melodies. These intriguing comparisons are presented in Figure 1(a). Transformer-based operator learners similarly apply global attention to inputs. As a result, physical locality remains underexploited in operator learning, leading to two major shortcomings: (i) Non-stationary information is largely encoded in local spatial features (Pacejka & Besselink, 1997). Since the temporal dimension is typically embedded implicitly during training, such non-stationarity is not explicitly represented (Barberi & Kruse, 2023). We therefore hypothesize that *once spatial locality is captured adequately, non-stationary temporal signals can be learned naturally.*

---

*Corresponding author.

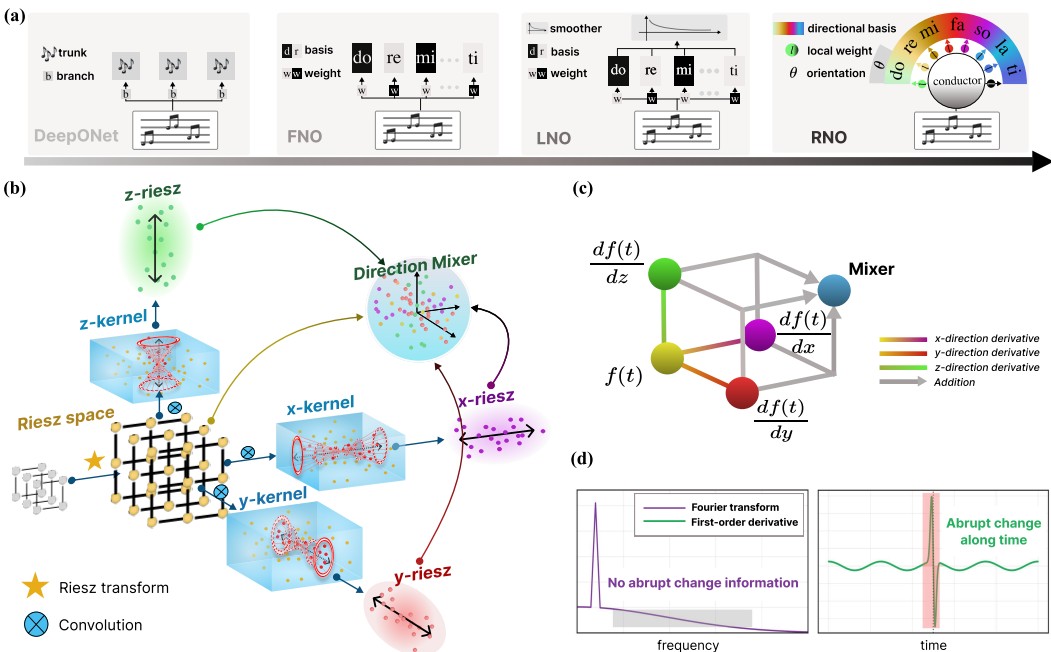

Figure 1: (a) Taking musical score processing as an example, we compare DeepONet, FNO, LNO, and RNO. (b) In the Riesz space, the Riesz transform yields direction-selective kernels (e.g., an *x*-kernel for the *x*-direction). Each kernel computes the spectral derivative along its associated direction. (c) The RNO pipeline is formalized mathematically. (d) In the Fourier spectrum (left), no localized change appears in the high-frequency band where a discontinuity is expected, whereas the first-order derivative (right) cleanly captures the emerging jump over time, demonstrating its local sensitivity.

To substantiate this hypothesis, we seek to reconstruct spatial locality more accurately and provide the proof in the A.3. (ii) Discarding spatial locality forfeits fine-scale details, especially the small perturbations typical of PDE-governed phenomena(Stearns, 1985; Xiong et al., 2025). From a spectral perspective, these correspond to high-frequency content; neglecting locality therefore suppresses high-frequency representations and weakens the modeling of strong nonlinearities as local detail naturally corresponds to high-frequency content in images(Wang et al., 2020). For further details on local necessity comparison, please refer to the test in A.1. In the experiments, phase plays a crucial role in the local spectral orientation. Although prior work addresses complex physical systems (Li et al., 2021; Wang et al., 2023), emphasizes local structure (Raonic et al., 2023; Wang et al., 2022) and uses derivative to optimize the loss(Cheng et al., 2025; Qiu et al., 2024), to our knowledge no operator-learning method that optimizes non-linear dynamics by mixing local derivatives. Concurrently, an increasing share of advances stems from incorporating established architectures (e.g., Transformer(Wu et al., 2023), mamba(Tiwari et al., 2025a), and graph network(Li et al., 2025), whereas recent attempts to modify the neural operator itself have produced limited gains. In these solver variants that introduce external structures (Liu et al., 2025; Kalimuthu et al., 2025), a key property of classical neural operators is disrupted: their ability to learn the intrinsic continuity of the underlying physical fields (Li et al., 2021). This disruption, in turn, limits the continuity of the model with respect to its inputs. Moreover, introducing external structures can partially compromise the efficiency advantages of classical neural operators.

Inspired by the Taylor expansion—where higher-order terms encode subtle, local variations (Kanwal & Liu, 1989; Bruhn et al., 2005), therefore derivatives are naturally suited to capture locality Figure 1(c). Yet operator-learning pipelines dominated by global transforms (e.g., Fourier, Laplace) often underrepresent such information, and the energy from abrupt changes is difficult to capture (Alley et al., 2003). To address these bottlenecks, we propose the Riesz Neural Operator (RNO), which improves neural operators' ability to model complex dynamic variations. The Riesz transform in our RNO acts as the **conductor**'s gestures, injecting directional derivatives that govern local dynamics, while the direction mixer serves as **orchestration**, harmonizing different directional

components. Analytically, Taylor expansion separates behavior: the DC (direct component) tracks slow trends, whereas abrupt or small-scale variations appear as higher-order terms (Hodaei et al., 2017; Bruhn et al., 2005). Building on this perspective, we apply the Riesz transform to extract first-order derivative information (direction and rate of change) (Unser & Van De Ville, 2009; Wadhwa et al., 2014; Tamada & Igarashi, 2017), and reproject these derivatives onto the original spectrum to emphasize salient components. The result is not only a faithful reproduction of the global score but also a nuanced capture of fine-scale, anisotropic variations — much like the layered expressiveness of a super symphony. The main contributions of this work are as follows:

- Motivated by the Taylor expansion, we encode fine-scale structure using local spectral derivatives. Integrating the Riesz transform sharpens sensitivity to dynamics while preserving physical interpretability, improving expressivity for non-linear phenomena.
- We are the first to explicitly embed the Riesz transform within operator learning, providing a direction-aware operator mixer to unify global basis information and local directional derivatives. Our method supports a principled composition for reconstructing physical dynamics.
- RNO achieves state-of-the-art performance in extensive benchmarks, covering various complex Navier–Stokes PDEs and the real-world climate ERA5 dataset. Experiments indicate that its low-level feature mixer affords a distinct advantage for modeling strong nonlinearities.

## 2 PRELIMINARIES

**Neural Operator.** Neural operator learning represents a novel paradigm intersecting scientific computing and machine learning, aimed at learning solution operators for PDEs. Owing to their full differentiability, neural operators permit direct parameter optimization, making them ideal for inverse design and a broad spectrum of inverse problem settings. Initially, neural operators originated from DeepONet(Lu et al., 2021). This paradigm later evolved into the classical FNO(Li et al., 2020a;b), which also performs function-to-function mappings in Fourier space and has spawned several FNO variants(Tran et al., 2021; Li et al., 2023). Recently, a study employed the Laplace transform as an alternative to the Fourier transform(Cao et al., 2024a) to address non-periodic characteristics in data, significantly enhancing the interpretability of neural operators. This highlights the significance of handling non-stationary data(Tiwari et al., 2025b). Additionally, other studies have concentrated on the capability of models to address complex phenomena(Li et al., 2021; Lanthaler et al., 2023; Xiao et al., 2024), strengthening the neural operators' ability to handle nonlinearity.

Meanwhile, due to the complex variability of real-world data and the higher standards required for capturing detailed variations, local properties within neural operators have also attracted attention(Raonic et al., 2023; Tripura & Chakraborty, 2022). Among these studies, the work on neural operators with localized integral and differential kernels emphasizes the necessity of local and differential properties. However, their differential kernels and local kernels are designed separately and introduced into the model as independent modules. Our approach leverages domain knowledge, integrates task-specific features with directional information, and thus possesses the capability to handle local signals, all while maintaining interpretability.

**Riesz Methods.** Spatial derivatives are widely used in deep learning. In Physics-Informed Neural Networks (PINNs) (Raissi et al., 2019), temporal/spatial derivatives construct PDE residuals, ensuring predictions both fit observations and satisfy governing laws. Derivative signals have likewise been incorporated into LSTM variants to better track state changes (DiPietro & Hager, 2020). By contrast, explicit frequency-domain operations are less common: most prior work analyzes networks' spectral behavior during training (Xu et al., 2019) rather than designing models centered on spectral directionality.

The Riesz transform maps spatial derivatives into the spectral domain while blending integral and differential effects: *the integral term preserves neural operator advantages, and the differential term injects local variation* (Tamada & Igarashi, 2017; Wadhwa et al., 2014). Here we use the vector Riesz transform from multidimensional signal processing, which generalizes the 1D Hilbert transform to higher dimensions; the relationship is derived in A.7. Formally, the transform is a convolution with a Calderón–Zygmund kernel (de Francia et al., 1986), and its kernel-based processing is illustrated in Figure 1(b). A defining property is strong directional selectivity: the transform decomposes signals

along specific spatial directions, enabling analysis of anisotropic structure (e.g., edges and textures) (Unser et al., 2009) and isolation of instantaneous phase. These properties make the Riesz transform well suited to data with complex, non-stationary variability.

# 3 METHODS

## 3.1 PROBLEM DEFINITION

In physics, it is widely recognized that PDEs characterize a function by constraining its local derivatives(Wu et al., 2023). That is, the evolution of a physical system is described through relations among its partial derivatives:

$$\mathcal{F}\big(x, u(x), \partial_{x_1} u(x), \ldots, \partial_{x_n} u(x)\big) = 0, \tag{1}$$

where $\partial_{x_i} u(x)$ denotes the local derivative of $u$ with respect to $x_i$. Physically, the presence of many local differential operators means that PDEs contain a wide range of higher-order dynamical quantities, which together govern the rich variety of behaviors observed in these systems. More broadly, accurately capturing small-scale variations is beneficial for physical signals. Since deep learning aims to exploit patterns, inputs with natural and approximately orthogonal dynamic components enhance the model's expressive capacity. In addition, many physical governing equations mirror optical-flow formulations that decompose fields into smooth backgrounds and dynamic perturbations; see A.9 for details. Nevertheless, the implications of this locality have been insufficiently exploited. We therefore seek a principled representation of this local derivative form. As the natural spectral representation of derivatives, the Riesz transform provides more than an engineering convenience: embedding it into neural operators offers a mathematically and physically consistent means of unifying global spectral efficiency with local dynamics. For a further relationship discussion of local derivative and PDEs with a much higher degree of mathematical rigor, we refer the readers to A.1.

## 3.2 DERIVATIVE INFORMATION IN PHYSICAL FIELDS

**Local dynamics via Taylor expansion.** Adopting the Eulerian viewpoint commonly used in optical flow analysis(Bruhn et al., 2005), we describe temporal variation of a one–dimensional signal by the first–order Taylor series

$$I(x,t) \approx f(x) + \gamma(t) \frac{df(x)}{dx}, \tag{2}$$

where $I(x,t) = f(x + \gamma(t))$ and $\gamma(t)$ denotes the instantaneous displacement at time $t > 0$ as shown in Figure 7. The term $\gamma(t) f'(x)$ therefore *encodes motion magnitude implicitly inside the spatial derivative*. Neglecting this factor, as in most vanilla neural operator blocks, effectively acts as a low-pass filter and can compromise fidelity in highly dynamic regimes.

**Directional derivative.** Beyond magnitude, first–order derivatives carry orientation information as described in A.1. The directional derivative of $u(\mathbf{x}, t)$ along a unit vector $\mathbf{n} = [n_1, \ldots, n_d]^\mathsf{T}$ is

$$\nabla u \cdot \mathbf{n} = \sum_{i=1}^{d} \frac{\partial u}{\partial x_i} n_i. \tag{3}$$

Consequently, model predictions must depend not only on scalar changes but also on the direction in which those changes occur.

**Spectral derivative.** Taking the temporal derivative of $f(t)$ and transferring it to the Riesz space gives

$$\gamma(t) \frac{df(t)}{dt} \quad \overset{\mathscr{R}}{\longleftrightarrow} \quad \gamma(t)\, j\, k\, \mathscr{R}\{f(t)\}, \tag{4}$$

where $k$ is the wave component and $\mathscr{R}$ denotes Riesz transform. In this representation, a spatial derivative manifests as a $90°$ phase shift accompanied by an amplitude scaling proportional to $k$. In 2D, $\mathscr{R}_x$ and $\mathscr{R}_y$ act as projectors onto directions defined by $k$'s angles with the axes, allowing efficient computation of spatial (and higher-order) derivatives. This relation enables the Riesz transform to compute spatial derivatives efficiently and to incorporate higher-order quantities.

**Dynamics in Riesz space.** Let $h(\mathbf{q}, t)$ be a scalar field with spatial coordinates $\mathbf{q} = [q_1, \ldots, q_n]^\mathsf{T} \in \mathbb{R}^n$ and time $t \in \mathbb{R}^+$. Its complex-valued spectrum is defined by the exponential kernel

$$\hat{h}(\mathbf{k}, t) = \int_{\mathbb{R}^n} h(\mathbf{q}, t) \, e^{-j\,\mathbf{kq}} \, d\mathbf{q}, \qquad \mathbf{k} = [k_1, \ldots, k_n]^\mathsf{T} \in \mathbb{C}^n, \tag{5}$$

where $j^2 = -1$. The magnitude $\|\mathbf{k}\| = (\sum_{i=1}^n k_i^2)^{1/2}$ controls spatial scale, while the vector direction of $\mathbf{k}$ encodes the dominant orientation of the corresponding spectral component. According to the definition of the directional derivative, we apply the $n$-D Riesz transform $\mathscr{R}$ to direction yields

$$\mathscr{R}\big[\nabla h(\mathbf{q}, t)\big](\mathbf{k}) = \sum_{i=1}^n j \, h_i \, \frac{k_i}{\|\mathbf{k}\|} \, h(\mathbf{q}, t) e^{-j\,\mathbf{kq}} d\mathbf{q}, \tag{6}$$

showing that differentiation becomes a multiplicative modulation by the directional factor $j \, h_i k_i / \|\mathbf{k}\|$. This highlights a key property of the Riesz framework: normalising by $\|\mathbf{k}\|$ preserves energy while amplifying orientation selectivity, enabling precise estimation of directional rate of change without additional scaling heuristics.

### 3.3 RIESZ NEURAL OPERATOR

**Operator learning.** To enhance representational capacity, the input is first projected into a higher-dimensional space by a shallow connection layer, yielding $q(t)$. The transformed $q(t)$ then passes through an integral-kernel layer and a linear layer, followed by an activation, to produce the high-dimensional output representation,

$$u(t) = \sigma\big(W q(t) + (c * q)(t)\big). \quad \forall t \in D \tag{7}$$

Here, $D$ is a bounded open set; $W$ is a linear transformation; and $(c * q)(t) := \int_D c(t, s) \, q(s) \, \mathrm{d}s$ denotes the integral operator with kernel $c$. By Green's function theory (Li et al., 2020b), the kernel $c(t, s)$ can approximate solution operators for PDEs. As shown in Figure 9, rather than relying on a single global transform, we couple local and global components to more fully map inputs to outputs. Indeed, RNO preserves the compositional structure characteristic of neural operators. Its overall procedure can be decomposed into three modules,

$$\mathscr{F}_\theta = \mathscr{F}_{\theta_{\mathrm{CoordToRiesz}}} \circ \mathscr{F}_{\sum_{i=1}^M \theta_i^R} \circ \mathscr{F}_{\theta_{\mathrm{RieszToCoord}}}, \tag{8}$$

where $\circ$ denotes operator composition and $\mathscr{F}$ is a mapping between spaces. Specifically, $\mathscr{F}_{\theta_{\mathrm{CoordToRiesz}}} : \mathcal{X} \to \mathcal{R}_\mathcal{X}$ lifts the input to the Riesz domain, and $\mathscr{F}_{\theta_{\mathrm{RieszToCoord}}} : \mathcal{R}_\mathcal{Y} \to \mathcal{Y}$ reconstructs the final output. This modular factorization is generic for neural operators. In the Riesz domain, $\mathscr{F}_{\sum_{i=1}^M \theta_i^R}$ learns and mixes non-linear, direction-wise mappings, where $M$ is the number of directional components. In practice, components are combined via analytic signals (1D) (Marple, 1999) or monogenic signals (multi-D) (Unser et al., 2009), providing a natural encoding of signal characteristics. Directional weights emphasize orientation-dependent variation. The enhanced Riesz representations are then mapped back by $\mathscr{F}_{\theta_{\mathrm{RieszToCoord}}}$ to yield a spatial output with enhanced directional features.

**Riesz conductor.** Analogous to convolution performed in a generic frequency domain, the kernel $(c * q)$ is realised by multiplication in Riesz space. For $m$-dimensional data the $i$-th directional Riesz transform of $q$ is

$$R_i(\mathbf{k}) = \int_{-\infty}^\infty \cdots \int_{-\infty}^\infty \frac{j \, k_i}{\|\mathbf{k}\|} \, e^{-j\,\mathbf{k}\cdot\mathbf{q}} \, d^m \mathbf{q}, \tag{9}$$

The spectral factor $j \, k_i / \|\mathbf{k}\|$ behaves as a scale-normalised first-order derivative, thus emphasising local variations while suppressing magnitude. Using the forward ($\mathscr{R}$) and inverse ($\mathscr{R}^{-1}$) Riesz transforms, the integral operator is

$$(c * q)(t) = \mathscr{R}^{-1}\big(\mathscr{R}(c) \cdot \mathscr{R}(q)\big)(t), \qquad \forall t \in D, \tag{10}$$

where "·" denotes element-wise complex multiplication. The model overview of RNO is shown in Figure 2. In the Riesz conductor, the directional numbers are not chosen randomly but are instead based on the data's dimensionality. This approach aligns with the inherent characteristics of physical field data. For example, in two-dimensional data, two orthogonal directions are used, maximizing

data utilization and optimizing results without introducing redundancy, the proof can be found in A.5. Each direction is also multiplied by a scaling factor $\zeta$, which controls the contribution of each direction. To account for the possibility that incorporating heterogeneous perturbations may introduce data artifacts, we establish an upper bound on $\zeta$,

$$\zeta \left( \sum_{i=1}^{d} \omega_i \gamma_i(t) \right) < \frac{\pi}{6}, \tag{11}$$

where $\omega_i$ denotes the frequency along the $i$-th direction. The upper bound on $\zeta$ follows from the aggregate total variation across directions; see A.6 for the full derivation. The resulting spectral enhancement is illustrated in Figure 2: the Riesz integral kernel exhibits increased energy in high-frequency detail bands, mitigating the high-frequency issues reported during training (Xu et al., 2019).

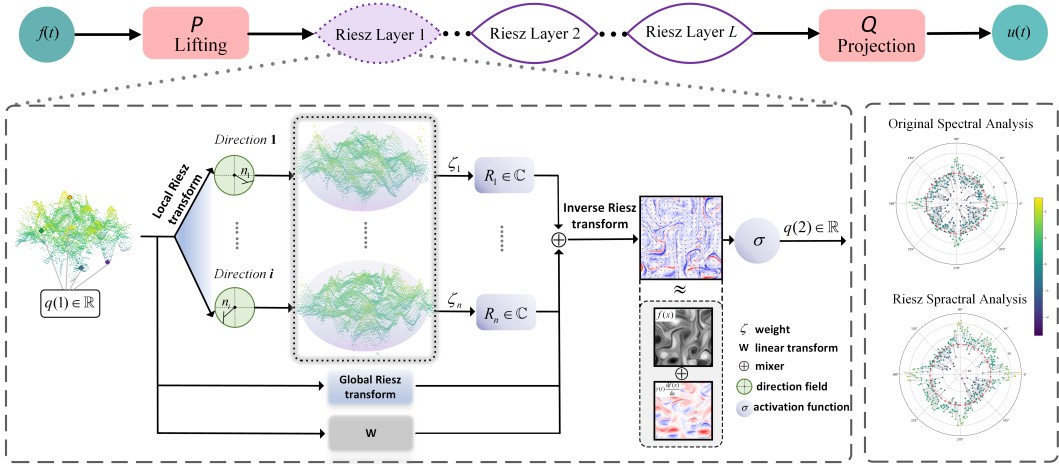

Figure 2: Model Overview of RNO. **Top:** The full RNO preserves the canonical neural-operator scaffold. **Bottom left:** Riesz layer 1. The input $q(1)$ is processed in parallel by (i) direction-wise local transforms, weighted by $\zeta$, to extract orthogonal local features $R$; (ii) a global transform that stabilizes the representation; (iii) a linear map $W$. The mixed output passes through an inverse Riesz transform to reinforce spatial locality, followed by an activation $\sigma$ to produce $q(2)$. **Bottom right:** Spectral comparison before and after the Riesz integral operator. Energy in the high-frequency band $f$ (red circle denotes spectral mean) is evaluated under a fixed reference: larger magnitudes appear as outward expansion, smaller as inward contraction.

**Local orchestration.** We introduce a direction-aware mixer in the Riesz domain that orchestrates global–local features across spectral scales. $\mathscr{F}_{\theta_{\mathrm{CoordToRiesz}}}$ aggregates *global* Riesz-spectrum statistics, while directional branches capture *local* anisotropic variation; their fusion yields a monogenic-style representation that preserves coherence and fine directional detail, beyond magnitude-only operators. Details of the construction are provided in A.10. Data layout and formulas appear in Figure 1. In 1-D, Cartesian channels are not meaningful, wavelength conditions mixing to fuse forward/backward components (implementation details are in A.11).

**How derivatives help.** Derivatives enhance the operator's physical representation by resolving local variations that global features cannot capture. As in a Taylor expansion, where derivative terms refine the leading field $u(x,t)$, they provide a principled way to encode fine-scale spatial and temporal changes. This yields a natural split: low-order global components model smooth, low-frequency structure, while orientation-aware derivatives concentrate energy at high wavenumbers and in anisotropic regions. In RNO, Riesz-based directional derivatives encode oriented changes in a latent spectral space, and a small set of mixing weights controls their contribution. During training, gradients emphasize directions that most reduce the loss, so the derivative channels supply missing detail while the global modes carry the coarse structure, leading to sharper and more physically meaningful predictions. For further discussion, see Sec. A.9.

Table 1: Test relative $\ell_2$ error ($\downarrow$) on five PDE benchmarks. Lower is better. The best (**bold**) and second-best (underline) results are highlighted. Promotions indicate relative gain of RNO over the second-best.

| Model | Relative $\ell_2$ ($\downarrow$) | | | | |
|---|---|---|---|---|---|
| | Duffing | Beam | Diffusion | Reaction–Diff. | Brusselator |
| U-NET(2015) | 0.3835 | 0.0522 | 0.0212 | 0.1049 | 0.2557 |
| GALERKIN(2021) | 0.4239 | 0.0971 | 0.0319 | 0.1983 | 0.1883 |
| DEEPONET(2021) | 0.6289 | 0.9790 | 0.1356 | 0.7969 | 0.1910 |
| FNO(2020a) | 0.4536 | 0.0821 | 0.0229 | 0.1214 | 0.1827 |
| WNO(2022) | 0.2532 | 0.0571 | 0.0179 | 0.1713 | 0.1789 |
| F-FNO(2023) | 0.2782 | 0.0515 | 0.0143 | 0.1003 | 0.1719 |
| LSM(2023) | 0.1699 | 0.0456 | 0.0145 | 0.0996 | 0.1628 |
| LNO(2024a) | 0.3325 | 0.0452 | 0.0081 | 0.1355 | 0.1858 |
| ONO(2024) | 0.3491 | 0.0519 | 0.0137 | 0.0989 | 0.1545 |
| **RNO** | **0.1663** | **0.0219** | **0.0079** | **0.0899** | **0.1317** |
| *Promotion* | 2.1% | 51.6% | 2.5% | 10.3% | 14.7% |

[*] Top 4 ranking methods of PDE benchmarks: RNO (ours), LNO (2024), ONO (2024), LSM (2023).

## 4 EXPERIMENTS

**Benchmark.** RNO is evaluated on five diverse PDE benchmarks (Cao et al., 2024a)—Duffing, Beam, Diffusion, Reaction–Diffusion, and Brusselator(spanning 1-D 3-D, details in A.18). To assess performance on complex and abrupt dynamics, we further test 2-D Navier Stokes equations at Reynolds numbers 40, 500, and 5000 (Li et al., 2021), where higher Reynolds numbers induce richer rotational structure, probing sensitivity to directional capture. We then compare RNO with classical baselines on the real-world ERA5 dataset (Hersbach et al., 2020). These tasks constitute canonical PDE application scenarios and reflect practical challenges in solving such equations.

**Baseline.** We compare RNO with seven well-established neural network models: a classic model, U-Net(Ronneberger et al., 2015); two transformer-based operator learning models, Galerkin Transformer(Cao, 2021) and LSM(Wu et al., 2023); and six classical and powerful neural operator models: DeepOnet(Lu et al., 2021), FNO(Li et al., 2020a), WNO(Tripura & Chakraborty, 2022), F-FNO(Tran et al., 2023), ONO(Xiao et al., 2024) and LNO(Cao et al., 2024a). Each model excels in its respective domain. The comparison enables a comprehensive evaluation of RNO's effectiveness.

**Implementation details.** To ensure fairness, we fix evaluation metrics and training epochs across methods and use Adam (Kingma & Ba, 2014) optimizer. Metrics: relative $\ell_2$ error for the general PDE and ERA5 benchmarks, and mean-squared error (MSE) for Navier Stokes equations. For more details, please refer to A.16.

### 4.1 MAIN RESULTS

**Performance across diverse PDEs.** To assess generality, we evaluate RNO on five canonical PDE systems: Duffing, Beam, Diffusion, Reaction diffusion, and Brusselator, spanning mechanical, diffusive, and reactive dynamics. RNO attains the lowest prediction error on all benchmarks (Table 1); on Beam it reduces error by 51.6% relative to the second-best method, This substantial margin underscores RNO's ability to capture structural response behavior with high precision. Similarly, on the Duffing system, a classical non-linear oscillator known for its chaotic dynamics, RNO achieves a 2.1% improvement, indicating its capacity to model sensitive dependence on initial conditions and inherent nonlinearity effectively.

On more diffusion-dominated tasks, such as Diffusion and Reaction-Diffusion(Reac-Diff.), RNO still outperforms prior approaches. While these systems are generally more stable and easier to approximate, RNO delivers 10.3% and 2.5% reductions in error, respectively, demonstrating its strength not only in complex non-linear settings but also in capturing fine-scale spatial dynamics

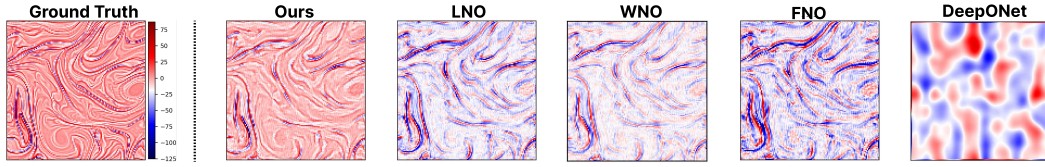

Figure 3: Comparison of high-frequency components with the Ground Truth.

Table 2: Mean-squared error (↓) results of RNO, LNO, WNO, F-FNO, FNO, and DeeepONet models on NS datasets.

| Datasets | DeepONet | FNO | F-FNO | WNO | LNO | RNO |
|---|---|---|---|---|---|---|
| $Re = 40$ | 0.0280 | 0.0078 | 0.0065 | 0.0077 | 0.0060 | **0.0049** |
| $Re = 500$ | 3.4082 | 1.4251 | 1.2312 | 0.9388 | 1.2117 | **0.4861** |
| $Re = 5000$ | 6.2721 | 2.9314 | 2.8912 | 2.5914 | 2.3139 | **0.9121** |

in smoother regimes. Finally, for the Brusselator, a prototypical model for nonlinear chemical oscillations and pattern formation, RNO achieves a 14.7% reduction in relative error. This result is particularly significant as it highlights RNO's capability to handle stiff systems with intricate feedback mechanisms and spatiotemporal instabilities.

**Performance on complex tasks.** To evaluate RNO under highly non-linear dynamics, we train on two-dimensional incompressible Navier Stokes snapshots at three Reynolds numbers $Re \in \{40, 500, 5000\}$. As $Re$ increases, the inertial range broadens and turbulent interactions intensify, rendering prediction progressively harder. Mean-squared error (MSE) for RNO, LNO, WNO, F-FNO, FNO and DeepONet is reported in Table 2. RNO achieves the lowest MSE across all $Re$; at $Re = 5000$ it attains MSE = 0.9121, substantially outperforming DeepONet, FNO, F-FNO, WNO and LNO. These results indicate that RNO's recurrent updates enhance stability as flow complexity grows. To assess fine-scale reconstruction, we isolate the high-frequency band of predicted velocity fields (Figure 3). RNO most faithfully recovers the high-wavenumber energy spectrum; LNO underestimates the spectral tail, whereas FNO exhibits ringing artefacts. DeepONet, however, produces a lot of noise in the high-frequency components. These findings confirm that RNO captures subtle non-linear fluctuations more accurately than the alternatives.

We further evaluate on ERA5 using hourly 850-hPa geopotential-height fields (2012–2022 for training; 2023 for testing). Further details of dataset can be found in A.18.7. As shown in Table 3, RNO significantly outperforms FNO , LNO, WNO, and DeepONet. RNO recovers both the inner low-frequency ring and the outer high-frequency core with smaller bias, as confirmed by PSD (Power spectral density) analysis (Figure 4). Across both synthetic turbulence and real-world weather, RNO preserves accuracy as dynamics intensify and faithfully reconstructs high-frequency content. It also transfers to operational-scale data without tuning, demonstrating its suitability for modeling and forecasting strongly non-linear spatiotemporal phenomena.

Table 3: Relative $\ell_2$ error (↓) results of RNO, LNO, WNO, FNO, and DeeepONet on ERA5.

| Dataset | DeepONet | FNO | WNO | LNO | RNO |
|---|---|---|---|---|---|
| ERA5 | 0.0912 | 0.0093 | 0.0085 | 0.0062 | **0.0022** |

## 4.2 MODEL ANALYSIS

**Ablation study.** To assess each module's contribution to the overall architecture, we evaluate four designs on two benchmarks (Duffing and Reaction–Diffusion): global-only (*g*) without mixer (*o*), local-only (*l*) without mixer, global+local (*g+l*) without mixer, and global+local (*g+l*) with mixer (*w*). From Table 4 we draw two clear observations: (i) on both datasets, either the global or the local module in isolation underperforms their combination; and (ii) the proposed mixer further facilitates effective information integration within the network, yielding additional gains.

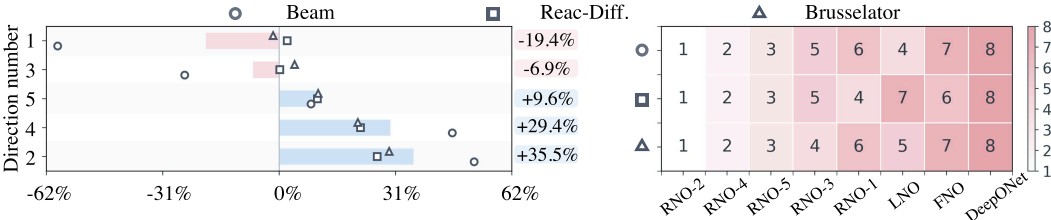

Figure 4: ERA5 benchmark at $t = 5$: Comparisons of 2-D PSD (top), spatial error (bottom right) and PSD along the $x$-axis for each method (bottom left).

**Influence of orthogonalization.** In practice, we set the number of directions equal to the data dimensionality to exploit the input's natural orthogonality. To isolate its effect, we vary the direction set and evaluate on three benchmarks (Beam, Reaction–Diffusion, and the Brusselator). We sweep the number of directions from 1 to 5, defining each by a uniform partition of the angular domain. We compare RNO with LNO, FNO, and DeepONet to quantify the role of orthogonal representations; in 2-D, two directions align with the native coordinate axes and are exactly orthogonal. Analogously in 3-D, the three canonical axes are orthogonal.

Figure 5: Comparison of relative $\ell_2$ errors (left) and rank (right) for RNO, LNO, FNO, and DeepONet on the Beam, Reaction–Diffusion, and Brusselator benchmarks as the direction number varies from 1 to 5. RNO-$n$ indicates that the RNO uses $n$ directions.

As shown in Figure 5, RNO achieves its best performance with exactly two directions and remains competitive(typically second-best), when the number of directions is an integer multiple of the data dimension (e.g., 4). With too few directions (e.g., 1), we observe information loss and occasional instability;

Table 4: Relative $\ell_2$ error ($\downarrow$) results of ablation study.

| Datasets | $g(o)$ | $l(o)$ | $g+l\,(o)$ | $g+l\,(w)$ |
|---|---|---|---|---|
| Duffing | 0.2098 | 0.2262 | 0.1801 | **0.1663** |
| Reac–Diff. | 0.0953 | 0.1189 | 0.0903 | **0.0899** |

with too many (e.g., 3 or 5), performance does not improve because the additional, non-orthogonal directions are redundant. Even under suboptimal counts, RNO outperforms LNO, FNO, and DeepONet on most metrics. This highlights that orthogonality is not just a heuristic choice, but a principled design element in neural operators.

**Influence of activation function.** In prior experiments, all methods employed the same $\sin$ activation to ensure parity. Prior work identifies activation functions as the primary source of nonlinearity in neural operators (Tran et al., 2023; Lu et al., 2021; Li et al., 2020b). To assess RNO's intrinsic capacity for non-linear phenomena, we benchmark it under multiple activations ($\sin$, GELU, ReLU, Leaky ReLU, Sigmoid, Tanh, and none) on two representative tasks: Duffing and Reaction–Diffusion.

As shown in Table 5, on Duffing RNO attains the lowest error under every activation; ReLU performs best, while $\sin$ is slightly suboptimal. Even with no activation, RNO surpasses FNO, LNO, and DeepONet by a large margin, indicating a stronger intrinsic capacity for non-linear dynamics. On Reaction diffusion, the pattern differs: with Sigmoid or Tanh, RNO no longer exceeds FNO or LNO. Removing the activation improves RNO, approaching the optimum in Table 5. We attribute this behavior to redundant nonlinearities introduced by Sigmoid/Tanh, which exceed the data's representational needs and destabilize gradient propagation. In contrast, eliminating the activation severely degrades LNO, FNO, and DeepONet, underscoring RNO's superior capacity to internalize nonlinear representations.

Table 5: Relative $\ell_2$ error ($\downarrow$) with different activation function on Duffing and Reac-Diffusion benchmarks. Lower is better. The best (**bold**) and second-best (underline) results are highlighted.

| Dataset | Model | Relative $\ell_2$ ($\downarrow$) | | | | | | |
|---------|-------|------|------|------|-----------|---------|------|------|
| | | sin | gelu | relu | leaky_relu | sigmoid | tanh | None |
| Duffing | **RNO** | **0.1663** | **0.1962** | **0.1441** | **0.1481** | **0.2089** | **0.1582** | **0.3215** |
| | LNO(2024a) | 0.3325 | 0.4010 | 0.4527 | 0.4828 | 0.4211 | 0.3662 | 0.8121 |
| | FNO(2020a) | 0.4536 | 0.4725 | 0.4555 | 0.4612 | 0.4913 | 0.4343 | 0.9296 |
| | DeepONet(2021) | 0.6289 | 0.5972 | 0.4278 | 0.4717 | 0.9993 | 0.7042 | 0.9787 |
| | *Promotion* | +50.0% | +51.1% | +68.2% | +67.9% | +50.4% | +56.8% | +60.4% |
| Reac–Diff. | **RNO** | **0.0899** | **0.1011** | **0.1027** | **0.0973** | 0.1323 | 0.1482 | **0.1017** |
| | LNO(2024a) | 0.1355 | 0.1277 | 0.1068 | 0.1111 | **0.1147** | **0.1069** | 0.2915 |
| | FNO(2020a) | 0.1214 | 0.1443 | 0.1337 | 0.1500 | 0.1322 | 0.1424 | 0.3511 |
| | DeepONet(2021) | 0.7969 | 0.2742 | 0.2739 | 0.2754 | 1.0000 | 0.4337 | 0.6967 |
| | *Promotion* | +25.9% | +20.8% | +3.8% | +12.4% | −15.3% | −38.6% | +65.1% |

**Efficiency analysis.** Beyond prediction accuracy, we also assess the computational efficiency of RNO against other neural operators. Figure 6 shows that RNO offers the best trade-off between error and training time on both Diffusion and Reaction–Diffusion. On Diffusion, it achieves the lowest error with a training time comparable to the fastest baselines. On Reaction–Diffusion, it remains both the most accurate and the fastest. Relative to the widely used FNO, RNO reduces the relative $\ell_2$ error by about $26\%$, trains nearly $2\times$ faster per epoch, and uses roughly half as many parameters. Despite its compact design, LNO is outperformed by RNO by more than $30\%$ in accuracy and over $5\times$ in training speed, showing that the gains do not rely on a larger model. DeepONet and WNO are both less accurate and slower, even with similar or larger parameter counts. Overall, RNO dominates the accuracy–efficiency–model-size trade-off, making it attractive for practical operator-learning applications (see Tables in A.12 for details).

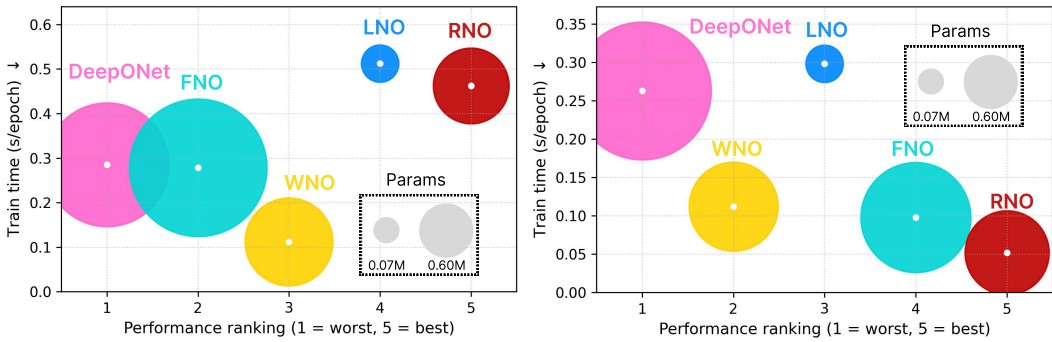

Figure 6: Efficiency comparison of all compared models on the benchmarks of Diffusion (left) and Reaction-Diff (right) datasets.

## CONCLUSION

We presented the Riesz Neural Operator (RNO), which bridges global spectral modeling with local derivative dynamics through Riesz transform. By jointly modeling local daymics (differentiation) and the operator theory (integration), we provide an integrated formulation that gives an intuitive explanation of how RNO enhances the interpretability. Evaluations on diverse PDEs, Navier–Stokes flows, and ERA5 data confirm consistent gains and robustness. Beyond accuracy, RNO demonstrates that embedding spectral derivatives into operator learning provides a natural and principled path toward more expressive models, suggesting broad potential for advancing scientific machine learning.

ACKNOWLEADGEMENTS

This work is financially supported by the National Natural Science Foundation of China (No. 12572266), National Key Research and Development Program (2024YFF1500600), Yongjiang Talent Program of Ningbo (No. 2022A-242-G), as well as by the High-Performance Computing Centers at Eastern Institute of Technology, Ningbo, and Ningbo Institute of Digital Twin.

ETHICS STATEMENT

We adhere to the ICLR Code of Ethics (https://iclr.cc/public/CodeOfEthics). This work involves no human participants, personally identifiable information, or sensitive content.

REPRODUCIBILITY STATEMENT

We provide detailed descriptions of benchmarks in A.18. Experimental settings are summarized in 4. The datasets used in this study are publicly available: the diverse PDE benchmarks at (https://github.com/qianyingcao/Laplace-Neural-Operator); Navier–Stokes with $Re \in \{40, 500, 5000\}$ at (https://github.com/neuraloperator/neuraloperator); and ERA5 at (https://cds.climate.copernicus.eu/). All results are averages over 3 runs with different random seeds. Definitions of the evaluation metrics are provided in A.17.

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

# Appendix Contents

Supplementary Material for RNO

# A APPENDIX

## A.1 WHY LOCAL DIRECTION IS IMPORTANT FOR PDEs?

Local derivatives are inherently directional, with directionality encoded by the phase gradient $\nabla\phi$. In the high-frequency regime, PDEs couple to the data through their principal symbol evaluated at $\nabla\phi$. The Riesz transform provides rotation-equivariant estimates of local phase and orientation, thereby supplying neural networks with precisely the quantities that govern PDE propagation. We next analyze each step in detail.

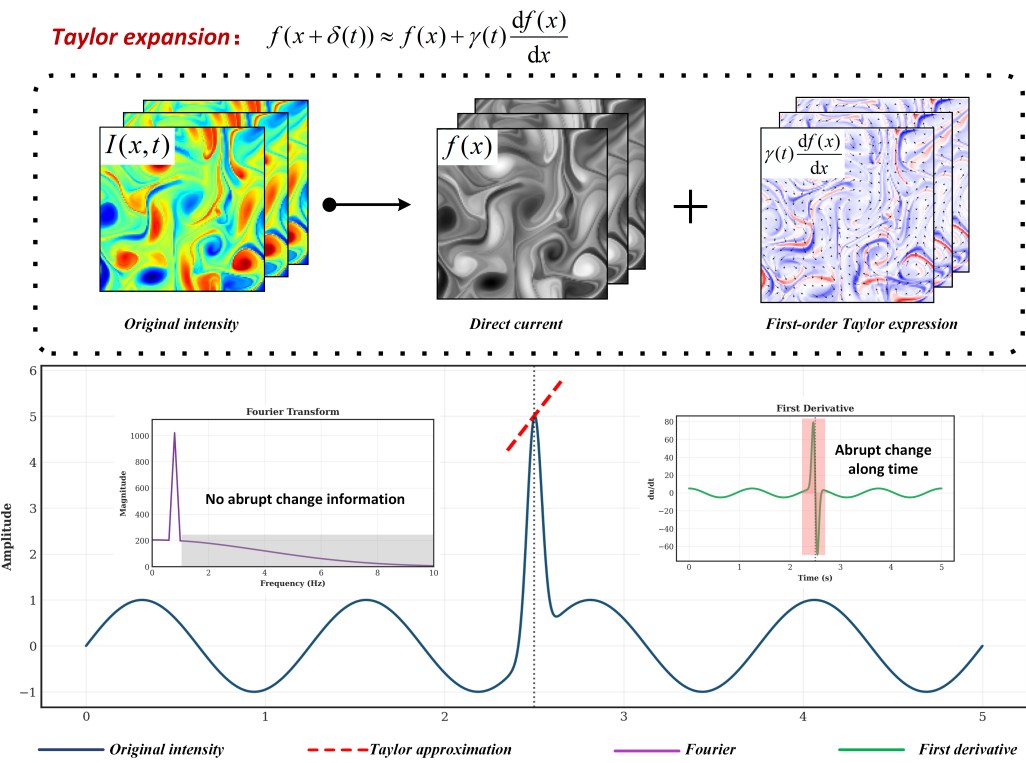

Figure 7: The specific structure of Taylor expansion and its difference from Fourier transform.

## 1. LOCAL DERIVATIVES ARE DIRECTIONAL

Given $u : \mathbb{R}^d \to \mathbb{C}$, the directional derivative along a unit vector $\mathbf{n}$ is

$$\partial_{\mathbf{n}}u(\mathbf{x}) = \mathbf{n}\cdot\nabla u(\mathbf{x}).$$

Write $u = Ae^{i\phi}$ with $A \geq 0$, $\phi \in \mathbb{R}$ wherever $u \neq 0$. Then

$$\partial_{\mathbf{n}}u = e^{i\phi}(\partial_{\mathbf{n}}A + iA\,\partial_{\mathbf{n}}\phi) = u\,(\partial_{\mathbf{n}}\ln A + i\,\partial_{\mathbf{n}}\phi). \tag{12}$$

Consequently, the normalized directional derivative decomposes neatly into amplitude and phase components:

$$\partial_{\mathbf{n}}\phi = \mathrm{Im}\Big(\frac{\partial_{\mathbf{n}}u}{u}\Big), \qquad \partial_{\mathbf{n}}\ln A = \mathrm{Re}\Big(\frac{\partial_{\mathbf{n}}u}{u}\Big). \tag{13}$$

Accordingly, local derivatives are inherently orientation dependent, with $\nabla\phi$ prescribing that orientation.

## 2. DIRECTIONALITY IS ENCODED BY THE PHASE

From $\nabla u = u(\nabla\ln A + i\,\nabla\phi)$ we obtain

$$\nabla\phi = \mathrm{Im}\Big(\frac{\nabla u}{u}\Big), \qquad \nabla\ln A = \mathrm{Re}\Big(\frac{\nabla u}{u}\Big). \tag{14}$$

The vector $\mathbf{k}(\mathbf{x}) := \nabla\phi(\mathbf{x})$ is the local wavevector and

$$\mathbf{n}(\mathbf{x}) := \frac{\nabla\phi(\mathbf{x})}{\|\nabla\phi(\mathbf{x})\|}$$

is the *local orientation*. In the high-frequency regime, where amplitude varies slowly compared with phase,

$$\|\nabla u\| \approx A\,\|\nabla\phi\|, \quad \partial_{\mathbf{n}}u \approx iA\,(\partial_{\mathbf{n}}\phi)\,e^{i\phi},$$

so the derivative is dominated by directional phase change.

## 3. WHY PDEs CARE: PRINCIPAL SYMBOL AND HIGH-FREQUENCY LINK

Consider an $m$-th order linear differential operator

$$P(x,\partial) = \sum_{|\alpha|\leq m} a_\alpha(x)\,\partial^\alpha, \qquad p_{\mathrm{pr}}(x,\xi) = \sum_{|\alpha|=m} a_\alpha(x)\,(i\xi)^\alpha$$

with principal symbol(Li et al.) $p_{\mathrm{pr}}$. Insert a ansatz $u(x) = A(x)e^{i\omega\phi(x)}$ ($\omega \gg 1$). By repeated Leibniz(Osler, 1970),

$$P\big(Ae^{i\omega\phi}\big) = e^{i\omega\phi}\Big[\omega^m\,p_{\mathrm{pr}}\big(x,\nabla\phi\big)\,A \,+\, \mathcal{O}(\omega^{m-1})\Big]. \tag{15}$$

At leading order, the principal symbol ties the dynamics to the phase gradient $\nabla\phi$. Asymptotic solvability at order $\omega^m$ entails the eikonal equation(Smith et al., 2020),

$$p_{\mathrm{pr}}\big(x,\nabla\phi(x)\big) = 0\,. \tag{16}$$

Hence, rays propagate along the phase gradient $\nabla\phi$. For the transport operator $P = \partial_t + \mathbf{a}(x)\cdot\nabla_x$, the principal symbol is $p_{\mathrm{pr}}(x,\tau,\xi) = i\big(\tau + \mathbf{a}\cdot\xi\big)$. Imposing the eikonal condition (Weinberg, 1962) $p_{\mathrm{pr}}(x,\partial_t\phi,\nabla_x\phi) = 0$ gives $\tau + \mathbf{a}\cdot\xi = 0$ with $\xi = \nabla_x\phi$, so the spatial propagation direction is aligned with $\nabla_x\phi$.

## 4. RIESZ TRANSFORM: EXTRACTING PHASE AND ORIENTATION

The $d$-D Riesz transform $\mathcal{R}u = (\mathcal{R}_1 u, \ldots, \mathcal{R}_d u)$ is defined by

$$\widehat{\mathcal{R}_j u}(\omega) = -i\,\frac{\omega_j}{\|\omega\|}\,\hat{u}(\omega).$$

**Plane wave benchmark.** For $u(x) = \cos(\mathbf{k}\cdot x)$,

$$\mathcal{R}u(x) = \sin(\mathbf{k}\cdot x)\,\frac{\mathbf{k}}{\|\mathbf{k}\|}.$$

Hence $\mathcal{R}u$ points in the wave direction $\mathbf{k}/\|\mathbf{k}\|$ and is $\pi/2$ out of phase.

**High frequency.** Let $u(x) = A(x)\cos\big(\omega\,\phi(x)\big)$ with smooth $A,\phi$ and $\omega \gg 1$. Locally near $x_0$, $\phi(x) \approx \phi(x_0) + \nabla\phi(x_0)\cdot(x - x_0)$, i.e. a plane wave with $\mathbf{k} = \omega\nabla\phi(x_0)$. Then

$$\mathcal{R}u(x_0) = A(x_0)\,\sin\big(\omega\phi(x_0)\big)\,\frac{\nabla\phi(x_0)}{\|\nabla\phi(x_0)\|} \,+\, \mathcal{O}\bigg(\frac{\|\nabla A\| + \|\nabla^2\phi\|}{\omega}\bigg). \tag{17}$$

Therefore the orientation is

$$\mathbf{n}(x) = \frac{\mathcal{R}u(x)}{\|\mathcal{R}u(x)\|} \approx \frac{\nabla\phi(x)}{\|\nabla\phi(x)\|}\,,$$

and a rotation-invariant *local phase* can be defined by the monogenic signal (Unser et al., 2009),

$$\rho(x) = \sqrt{u(x)^2 + \|\mathcal{R}u(x)\|^2}, \qquad \theta(x) = \mathrm{atan2}\big(\|\mathcal{R}u(x)\|,\,u(x)\big),$$

so that locally $u \approx \rho\cos\theta$, $\|\mathcal{R}u\| \approx \rho\sin\theta$.

## 5. THE RATIONALITY OF USING RIESZ TRANSFORMATION FOR NEURAL OPERATOR

- **Aligns with principal symbols.** By equation 15–equation 16, the leading-order response of the PDE depends on $\nabla\phi$ through $p_{\mathrm{pr}}(x, \nabla\phi)$. Riesz features extract the orientation $\mathbf{n} \approx \nabla\phi/\|\nabla\phi\|$ and the quadrature phase $\theta$, thereby granting the network direct access to the directional variables relevant to the PDE.
- **Rotation covariance.** Since $\mathcal{R}$ is rotation-covariant, $(u, \mathcal{R}u)$ provides features that transform predictably under rotations. This property is essential for modeling anisotropic diffusion, advection, and wave propagation without relearning all orientations.
- **Phase emphasis and contrast robustness.** Phase (via $\theta$) remains invariant under multiplicative contrast changes $u \mapsto c\,u$, while the derivatives in equation 13 accentuate $\partial_{\mathbf{n}}\phi$. Consequently, learning is guided toward geometry and propagation rather than raw amplitude.
- **Differentiable and lightweight.** $\mathcal{R}$ is linear with a fixed multiplier $-i\xi/\|\xi\|$, and its discrete realizations are convolutional and backpropagation-friendly. It can serve either as a fixed front-end or as a parallel stream complementing learned filters.

## 6. INFLUENCE OF PHASE AND ORIENTATION

To highlight how phase governs directionality, we conduct an ablation on the Navier–Stokes dataset. Under an identical network architecture, we compare two input parameterizations (cf. Figure 8(a)): (i) complex features that retain both magnitude and phase, $A(x)e^{i\phi(x)}$; and (ii) magnitude-only features with the phase suppressed, $A(x)e^{i*0}$. All other training and evaluation settings are held fixed.

As shown in Figure 8(b)-(d), removing phase yields markedly sparser fine-scale structures and substantial misalignment of local feature orientations. In rollout regimes (Cao et al., 2024b) (e.g., NO), such local directional errors accumulate over time. The effect is amplified for more complex data with more numerous localized structures, resulting in larger long-horizon errors, as evidenced by Table 2. Moreover, the Riesz transform precisely enhances the directional sensitivity of phase control.

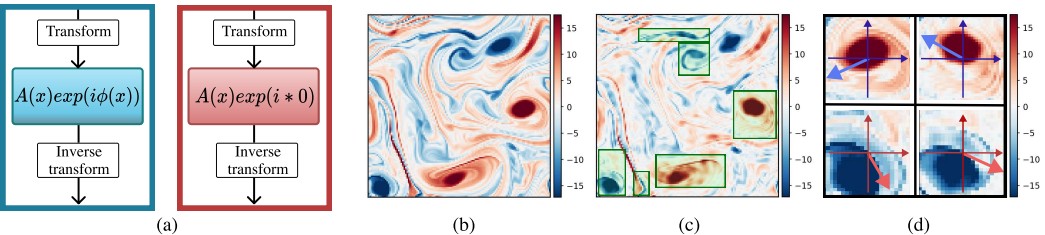

Figure 8: (a) Two feature structures; (b) results using magnitude–phase features; (c) results using magnitude-only features; (d) orientation comparison on selected details.

### A.2 FIRST−ORDER APPROXIMATION.

Assume that the image profile $f$ is twice continuously differentiable and that the displacement $\gamma(t)$ is small (Wadhwa et al., 2014). Since

$$I(x,t) = f\big(x + \gamma(t)\big),$$

we expand $f$ at position $x$ using Taylor's theorem with remainder:

$$f\big(x + \gamma(t)\big) = f(x) + \gamma(t)f'(x) + \frac{\gamma(t)^2}{2}\, f''(\xi(x,t)),$$

for some $\xi(x,t)$ between $x$ and $x + \gamma(t)$. If $|f''(z)| \le M$ in the relevant neighborhood, then the second–order remainder satisfies

$$\left|\frac{\gamma(t)^2}{2}\, f''(\xi(x,t))\right| \le \frac{M}{2}\,|\gamma(t)|^2.$$

Hence, whenever $|\gamma(t)|$ is small, the remainder is of order $|\gamma(t)|^2$, which is negligible compared to the first–order term of order $|\gamma(t)|$. Therefore we obtain the first–order approximation

$$I(x,t) = f\big(x + \gamma(t)\big) \approx f(x) + \gamma(t)\,\frac{df(x)}{dx}.$$

Intuitively, this means that for sufficiently small displacements the change in intensity is well described by a linear response with slope $f'(x)$, while higher–order (curvature) effects only appear at quadratic (and smaller) scales in $\gamma(t)$.

## A.3   PROPOSITION: PLUG-IN STABILITY FOR TEMPORAL LEARNING FROM SPATIAL LOCALITY.

Let $u : \Omega \times [0, T] \to \mathbb{R}$ solve a local evolution law of the form

$$\partial_t u(x,t) \;=\; F\big(x, t, \Psi(u)(x,t)\big), \qquad \Psi(u) := \big(u, \nabla u, \ldots, \nabla^m u\big), \tag{18}$$

on a bounded domain $\Omega \subset \mathbb{R}^d$, where $F$ is uniformly *locally Lipschitz* in its last argument: there exists $L > 0$ such that

$$\big|F(x,t,z) - F(x,t,z')\big| \;\le\; L\,\|z - z'\| \quad \text{for all } (x,t) \in \Omega \times [0,T],\ z, z' \in \mathbb{R}^q. \tag{19}$$

Assume a learned *spatial locality representation* $r(x,t)$ approximates $\Psi(u)(x,t)$ with

$$\big\| r - \Psi(u) \big\|_{L^p(\Omega \times [0,T])} \;\le\; \varepsilon, \qquad 1 \le p \le \infty. \tag{20}$$

Then for any $\delta > 0$ there exists a (pointwise) network $g_\theta : \mathbb{R}^{d+2+q} \to \mathbb{R}$ such that

$$\big\| g_\theta(x,t,r) - F\big(x,t,\Psi(u)\big) \big\|_{L^p(\Omega \times [0,T])} \;\le\; L\varepsilon \;+\; \delta. \tag{21}$$

*Proof (sketch).* Define $G(x,t,z) := F(x,t,z)$. By the Lipschitz property above,

$$\big| G(x,t,r) - G\big(x,t,\Psi(u)\big) \big| \;\le\; L\,\|r - \Psi(u)\|.$$

Taking $L^p$ norms yields the term $L\varepsilon$ by the assumption on $r$. Continuity of $G$ on compact subsets implies, by universal approximation, the existence of a small MLP $g_\theta$ such that $\sup_{(x,t,z) \in K} |g_\theta(x,t,z) - G(x,t,z)| \le \delta$ on the relevant compact set $K$. A triangle inequality then gives the stated bound.

In regimes where

$$\partial_t u(x,t) \;\approx\; \alpha(x,t)\,\big(n(x,t)\cdot\nabla u(x,t)\big), \tag{22}$$

with orientation $n(x,t) \in \mathbb{S}^{d-1}$ and scalar speed $\alpha(x,t)$, let $\varphi := n\cdot\nabla u$ and suppose $\|\widehat{\varphi} - \varphi\|_{L^p} \le \varepsilon$. If a small regressor $a_\theta(x,t,\cdot)$ estimates $\alpha$ with $\|\widehat{\alpha} - \alpha\|_{L^\infty} \le \eta$, then

$$\big\|\partial_t u - \widehat{\alpha}\,\widehat{\varphi}\big\|_{L^p} \;\le\; \|\alpha\|_{L^\infty}\,\varepsilon \;+\; \|\widehat{\varphi}\|_{L^\infty}\,\eta. \tag{23}$$

Thus, temporal accuracy follows directly from the fidelity of spatial directional derivatives and a scalar modulation fit.

**Remark (why the RNO representation satisfies the assumptions).**   The Riesz-based features $Ru = (R_1 u, \ldots, R_d u)$, with multipliers $\widehat{R_i u}(\xi) = -\mathrm{i}\,\xi_i/\|\xi\|\,\widehat{u}(\xi)$, form an energy-preserving system: $\sum_{i=1}^d \|R_i u\|_{L^2}^2 = \|u\|_{L^2}^2$ (Plancherel). Hence $Ru$ provides a stable, grid-agnostic surrogate of spatial locality (values, gradients, orientations). Once $r \approx \Psi(u)$ (or $r \approx (u, Ru)$) holds to accuracy $\varepsilon$, Proposition (P4) shows that the *non-stationary* temporal signal $\partial_t u$ can be learned by a low-capacity pointwise map with error bounded by $L\varepsilon + \delta$. In short: *once spatial locality is captured adequately, temporal non-stationarity becomes a stable plug-in regression problem.*

## A.4   GREEN FUNCTION AND RNO

We recall that many linear (or linearized) PDEs admit a Green representation (Li et al., 2020b) $Lu = f \Rightarrow u(x) = \int_\Omega G(x,y)\,f(y)\,\mathrm{d}y$, so the solution operator $\mathcal{G} : f \mapsto u$ is an integral operator with kernel $G$. Neural operators approximate such maps by learning an integral kernel $\kappa_\theta$ so that

$$(\mathcal{N}_\theta f)(x) = \int_\Omega \kappa_\theta(x,y)\,f(y)\,\mathrm{d}y \approx \int_\Omega G(x,y)\,f(y)\,\mathrm{d}y.$$

In Fourier Neural Operators, one further assumes $\kappa_\theta(x,y) = k_\theta(x-y)$ and parameterizes its Fourier transform by a multiplier $M_\theta(\xi)$; the operator becomes $(\mathcal{N}_\theta f)\hat{}\,(\xi) = M_\theta(\xi)\,\hat{f}(\xi)$, which is still

a bounded convolution operator with a (learned) Green–type kernel $k_\theta$ that is independent of the discretization.::contentReference[oaicite:0]index=0

RNO augments this construction with Riesz–based directional derivatives while preserving the same operator class. The $i$th Riesz transform can be written as a singular integral $R_i f = r_i * f$ with kernel $r_i(x) \sim x_i/|x|^{d+1}$ and Fourier multiplier $\widehat{R_i f}(\xi) = m_i(\xi)\,\hat{f}(\xi)$, where $m_i(\xi) = -\mathrm{i}\,\xi_i/\|\xi\|$ is bounded and homogeneous of order $0$. Thus $R_i$ is a Calderón–Zygmund operator (Journé, 1985), bounded on $L^2$ and on Sobolev spaces (Adams & Fournier, 2003), and does not destroy continuity of the solution operator. In RNO we apply the neural operator to the augmented field $q = f + \sum_i w_i R_i f$, so that

$$(\mathcal{N}_\theta^{\mathrm{RNO}} f)(x) = \int_\Omega \kappa_\theta(x,y)\,q(y)\,\mathrm{d}y = \int_\Omega \kappa_\theta(x,y) f(y)\,\mathrm{d}y + \sum_i w_i \int_\Omega \kappa_\theta(x,y)\,(R_i f)(y)\,\mathrm{d}y.$$

Writing $R_i f = r_i * f$ and exchanging integrals gives

$$(\mathcal{N}_\theta^{\mathrm{RNO}} f)(x) = \int_\Omega \widetilde{G}_\theta(x,z)\,f(z)\,\mathrm{d}z, \qquad \widetilde{G}_\theta(x,z) = k_\theta(x-z) + \sum_i w_i \int_\Omega k_\theta(x-y)\,r_i(y-z)\,\mathrm{d}y,$$

so $\mathcal{N}_\theta^{\mathrm{RNO}}$ is again an integral operator with a learned kernel $\widetilde{G}_\theta$ of Green–function type. In the Fourier domain, this is simply a new bounded multiplier $\widetilde{M}_\theta(\xi) = M_\theta(\xi)\big(1 + \sum_i w_i m_i(\xi)\big)$. Hence the Riesz term acts as a directional, high–frequency reweighting that injects local anisotropic variation into the representation, while the overall map remains a continuous, resolution–invariant integral operator in the same functional class as the original Green–kernel construction.

## A.5   ORTHOGONALITY

**Inner product.**   Let $n \geq 2$ and wave-vector $\mathbf{k} = (k_1,\ldots,k_n) \in \mathbb{R}^n$ and magnitude $\|\mathbf{k}\| = (\sum_{i=1}^n k_i^2)^{1/2}$. For any distinct indices $i \neq j$ consider the $L_2$ inner product

$$\langle R_i, R_j \rangle \;=\; \int_{\mathbb{R}^n} R_i(\mathbf{k})\,\overline{R_j(\mathbf{k})}\,d\mathbf{k}. \tag{24}$$

Then setting $E(\mathbf{k}) := |F(\mathbf{k})|^2$ (an even function in every $k_\ell$) yields

$$\langle R_i, R_j \rangle \;=\; \int_{\mathbb{R}^n} \frac{k_i k_j}{\|\mathbf{k}\|^2}\,E(\mathbf{k})\,d\mathbf{k}.$$

**Odd–even symmetry argument.**   Fix $i \neq j$. Under the sign change $k_i \mapsto -k_i$ the factor $k_i k_j$ flips sign while both $\|\mathbf{k}\|^2$ and $E(\mathbf{k})$ remain unchanged; hence the integrand is *odd* with respect to the $k_i$ axis. Because the integration domain $\mathbb{R}^n$ is symmetric, the integral vanishes:

$$\langle \mathcal{R}_i, \mathcal{R}_j \rangle = 0, \qquad \forall\, i \neq j \tag{25}$$

**Spatial-domain equivalence.**   By Parseval's theorem (Orfanidis, 1995), the frequency-domain result implies spatial orthogonality: $\langle \mathcal{R}_i f, \mathcal{R}_j f \rangle_{\mathbf{x}} = 0$ for $i \neq j$, where $\mathcal{R}_i$ denotes the real-space Riesz transform in direction $i$. Thus the vector of Riesz components $\mathcal{R} f = (\mathcal{R}_1 f, \ldots, \mathcal{R}_d f)$ forms an orthonormal frame in $L_2(\mathbb{R}^n)$. Consequently, neural operators that process the Riesz vector component-wise inherit stable, non-redundant latent representations in any spatial dimension $d$.

## A.6   BOUND

**First-order approximation.**   The first-order Taylor expansion provides an approximation to the actual intensity function $f(x)$ and its variations, which is reflected in the Riesz transform within the spatial domain (Wadhwa et al., 2014). For rapidly varying data, the accuracy of the first-order Taylor expansion decreases as $\zeta$ increases. Large values of $\zeta\gamma(t)$ may introduce artifacts. To ensure the approximation remains valid, we derive an upper bound for $\zeta$.

Assuming the first-order Taylor expansion holds, the modified intensity is:

$$f(x + \zeta\gamma(t)) \approx f(x) + \zeta\gamma(t)\frac{df(x)}{dx}. \tag{26}$$

The Riesz transform introduces a 90-degree phase shift. If $f(x) = \cos(\omega x)$, we have:

$$\cos(\omega x + \zeta \omega \gamma(t)) \approx \cos(\omega x) - \zeta \omega \gamma(t) \sin(\omega x). \tag{27}$$

Using the trigonometric identity:

$$\cos(\omega x + \zeta \omega \gamma(t)) = \cos(\omega x) \cos(\zeta \omega \gamma(t)) - \sin(\omega x) \sin(\zeta \omega \gamma(t)), \tag{28}$$

we require $\zeta \omega \gamma(t)$ to be small for $\cos(\zeta \omega \gamma(t)) \approx 1$ and $\sin(\zeta \omega \gamma(t)) \approx \zeta \omega \gamma(t)$. Assuming a 5% error tolerance, the second-order term should be less than 5% of the first-order term, yielding the condition:

$$\zeta \gamma(t) < \frac{\lambda}{12}, \tag{29}$$

where $\lambda = \frac{2\pi}{\omega}$ is the spatial wavelength. A larger wavelength allows for a larger scaling factor $\zeta$; smaller wavelengths require a smaller $\zeta$ to avoid artifacts. While this bound on $\zeta$ is theoretical, the wavelength of the data must be determined based on the signal and network's characteristics.

**Multidimensional case.** For a multi-dimensional setting, let $x = (x_1, x_2, \ldots, x_d)$ represent the spatial coordinates. The first-order Taylor expansion of $f(x)$ becomes:

$$f(x + \zeta \gamma(t)) \approx f(x) + \zeta \sum_{i=1}^{d} \gamma_i(t) \frac{\partial f(x)}{\partial x_i}. \tag{30}$$

Here, $\gamma_i(t)$ is the scaling factor in the $i$-th direction. The scaling factor $\zeta$ still amplifies the signal's variations across all dimensions. The error condition becomes:

$$\zeta \left( \sum_{i=1}^{d} \omega_i \gamma_i(t) \right) < \frac{\pi}{6}, \tag{31}$$

where $\omega_i$ is the frequency in the $i$-th direction. The bound provided here is not an exact limit but rather a theoretical upper bound. Since it depends on the angular frequency of the data, the boundary is not fixed. Accordingly, in the mixer A.10 we replace it with a learnable parameter while constraining its range.

### A.7 RELATIONSHIP BETWEEN THE HILBERT AND RIESZ TRANSFORMS

In one dimension, the Hilbert transform $\mathcal{H}f$ (Cizek, 1970) is the singular integral

$$\mathcal{H}f(x) = \frac{1}{\pi} \, \text{p.v.} \int_{-\infty}^{\infty} \frac{f(y)}{x - y} \, dy,$$

with Fourier representation

$$\mathcal{F}[\mathcal{H}f](\xi) = -i \, \text{sgn}(\xi) \, \hat{f}(\xi),$$

where "p.v." denotes the Cauchy principal value (Guiggiani & Casalini, 1987). In $\mathbb{R}^n$, the Riesz transform generalizes this construction: for $f \in \mathcal{S}(\mathbb{R}^n)$,

$$\mathcal{R}_j f(x) = \mathcal{F}^{-1} \left[ -i \frac{\xi_j}{|\xi|} \hat{f}(\xi) \right](x), \qquad j = 1, \ldots, n,$$

so each $\mathcal{R}_j$ acts as a normalized directional derivative with a degree-zero Fourier multiplier.

For $n = 1$, taking $\xi_1 = \xi$ yields

$$\mathcal{R}_1 f(x) = \mathcal{F}^{-1} \left[ -i \frac{\xi}{|\xi|} \hat{f}(\xi) \right](x) = \mathcal{F}^{-1} \left[ -i \, \text{sgn}(\xi) \, \hat{f}(\xi) \right](x),$$

recovering exactly the Hilbert transform: $\mathcal{R}_1 f = \mathcal{H}f$.

Collecting the components $\mathcal{R}f = (\mathcal{R}_1 f, \ldots, \mathcal{R}_n f)$ gives a vector operator that is an $L^2$-isometry:

$$\sum_{j=1}^{n} \|\mathcal{R}_j f\|_{L^2}^2 = \|f\|_{L^2}^2.$$

Thus the Riesz transform is the canonical multi-dimensional analogue of the Hilbert transform, widely used to extract directional and phase information in higher-dimensional signal analysis. Making this connection explicit sharpens our understanding of how Riesz transforms operate on derivative-like quantities and underpins the RNO framework.

Figure 9: A comparison between RNO and simplified versions of some classical architectures, highlighting the global-local mixer of RNO.

### A.8 DIFFERENCES BETWEEN RNO AND FNO + CNN

**Resolution invariance and local–global representation.** A key advantage of the Fourier Neural Operator (FNO) is *resolution invariance*: it approximates a continuous operator $\mathcal{T} : L^2(\Omega) \to L^2(\Omega)$ via a grid-agnostic spectral multiplier. Writing one FNO layer as

$$u_{l+1} = \sigma\Big(\mathcal{F}^{-1}\big[M(\xi)\,\widehat{u}_l(\xi)\big] + b\Big), \tag{32}$$

the learned multiplier $M(\xi)$ is evaluated on the Fourier samples of any discretization. Equivalently, the sampling map $S_h$ approximately commutes with $\mathcal{T}_M$:

$$S_{h'}\,\mathcal{T}_M \approx \mathcal{T}_M^{(h')}\,S_h, \tag{33}$$

so training and inference can use different resolutions with minimal drift (apply, then sample $\approx$ sample, then apply).

By contrast, FNO+CNN (Kalimuthu et al., 2025; Liu et al., 2025) augments equation 32 with a spatial convolution $(\kappa * u)(x)$ whose kernel $\kappa \in \mathbb{R}^{k \times k}$ is defined in pixel units. Under a grid change $h \to h'$, the effective continuous kernel rescales $\kappa_{h'}(x) = \kappa(x/h')$, hence $\widehat{\kappa_{h'}}(\xi)$ depends on $h'$. This breaks equation 33: the conv branch is not grid-agnostic, making cross-resolution generalization fragile.

We retain the operator view and expose *directional* locality spectrally via the Riesz transform:

$$\widehat{R_i u}(\xi) = -\mathrm{i}\,\frac{\xi_i}{\|\xi\|}\,\widehat{u}(\xi), \qquad \xi \neq 0, \tag{34}$$

a scale-normalized directional derivative. Intuitively, the spectral path preserves continuity across grids, while Riesz channels add anisotropic, high-wavenumber detail that complements global low-frequency modes.

**Complexity and redundancy.** Let $N = |\Omega_h|$ denote grid points and $C_{\text{in}}, C_{\text{out}}$ channel counts. An FNO layer with $|\Lambda|$ retained modes costs

$$\underbrace{\mathcal{O}\big(N \log N \cdot C_{\text{in}}\big)}_{\text{FFT/IFFT}} + \underbrace{\mathcal{O}\big(|\Lambda| \cdot C_{\text{in}} C_{\text{out}}\big)}_{\text{mode mixing}} .$$

Adding a $k \times k$ CNN (LeCun & Bengio, 1998) head introduces

$$\underbrace{\mathcal{O}\big(k^2\,N\,C_{\text{in}} C_{\text{out}}\big)}_{\text{local conv}},$$

Table 6: Comparison of model effiency on the Reaction–Diffusion dataset.

| Model | Params | Train time (s/epoch) ↓ | Peak memory (MB) ↓ | Relative $\ell_2$ ↓ |
|---|---|---|---|---|
| E-FNO | 793985 | 0.166 | 241.07 | 0.1131 |
| loglo-FNO | 764225 | 0.142 | 617.22 | 0.1009 |
| **RNO (ours)** | **172229** | **0.052** | **111.17** | **0.0899** |

plus feature-map memory $\propto N \times C$. For typical $k \in [3, 7]$ and moderate $|\Lambda|$, this CNN cost dominates as resolution grows, and features remain resolution-bound. Moreover, since filter responses $H_m(\xi)$ are unconstrained, concatenating CNN and spectral features often duplicates energy in overlapping bands, inflating rank unless explicit decorrelation is enforced.

In contrast, adding $d$ Riesz channels equation 34 costs

$$\underbrace{\mathcal{O}(N \log N \cdot C_{\text{in}})}_{\text{shared RT/IRT}} + \underbrace{\mathcal{O}(d\,N\,C_{\text{in}})}_{\text{elementwise multipliers}} + \underbrace{\mathcal{O}(C_{\text{in}}C_{\text{out}})}_{\text{light mixer}},$$

with $d = \dim(\Omega) \in \{2, 3\}$. The overhead is linear in $d$ and grid-agnostic; no extra $k^2 N$ spatial conv is introduced, and memory adds only $d$ feature maps plus a small mixer. The experimental results are shown in the 6, RNO achieves faster speed and higher accuracy.

**Why Riesz features are energy-orthogonal, while CNN features can be redundant.** By Plancherel (Delorme, 1998),

$$\sum_{i=1}^{d} \|R_i u\|_{L^2}^2 = \int_{\mathbb{R}^d} \Big( \sum_{i=1}^{d} \frac{\xi_i^2}{\|\xi\|^2} \Big) |\widehat{u}(\xi)|^2 \, \mathrm{d}\xi = \|u\|_{L^2}^2,$$

so the vector Riesz transform $Ru = (R_1 u, \dots, R_d u)$ is an $L^2$ isometry: it redistributes the same energy across $d$ directions without gain or loss. On band-limited discrete grids, the identity holds up to negligible aliasing, and channels are approximately decorrelated for isotropic content. Intuitively, we *split* energy by orientation rather than duplicating it.

For a CNN head with filters $\{\kappa_m\}_{m=1}^{C'}$ and responses $H_m(\xi)$, the feature energy at $\xi$ scales as $\sum_m |H_m(\xi)|^2 |\widehat{u}(\xi)|^2$. Without spectral constraints, multiple filters can attend the same bands; the Gram matrix $G_{mn} = \langle \kappa_m, \kappa_n \rangle$ is non-diagonal, yielding correlated channels and effective rank inflation. Our orthogonality study (Fig. 4.2) and analysis in App. A.5 support this view.

A.9 DIRECTIONAL DERIVATIVES AND FULL-SPECTRUM MODELING

As discussed in 3.3, derivative features are central to how RNO couples global and local dynamics. We now provide a concise, implementation-level description that links full-spectrum modeling, rank reduction, and learning dynamics.

**Representation.** In the spectral formulation, the $i$-th Riesz channel is obtained by the fixed multiplier $m_i(k) = -\mathrm{i}\, k_i/(\|k\|)$, implemented via FFT as discussed in A.16.1. The resulting vector of channels $\{R_i(q)\}_{i=1}^{d}$ behaves as a normalized first-order directional derivative. In the continuous limit, the transform is energy preserving and its components are orthogonal; on discrete, band-limited grids this property holds approximately. Intuitively, the global integral kernel captures smooth structure $I(x,t)$2, whereas anisotropic variations concentrate in the Riesz channels.Together they provide a compact, non-redundant coverage of the full spectrum.

**PDE and optical flow.** PDEs characterize a field by constraining its local derivatives (Wu et al., 2023):

$$\mathcal{F}\big(x, t,\ u(x,t),\ \partial_t u(x,t),\ \nabla u(x,t), \dots \big) = 0, \tag{35}$$

where $\partial_{x_i} u$ denotes the derivative of $u$ w.r.t. $x_i$. While derivatives quantify space–time variations, the induced solution operators may couple information globally or scale-locally. Hence, a useful representation should make both global modes and orientation-aware local changes explicit, rather than assuming pure locality.

A concise local view is the optical-flow–type (Beauchemin & Barron, 1995) transport along characteristics,

$$\partial_t u(x,t) + \mathbf{v}(x,t) \cdot \nabla u(x,t) \ \approx \ S(u,x,t) + \nu\,\Delta u(x,t), \tag{36}$$

whose brightness-constancy case is $S \equiv 0$, $\nu = 0$. This form cleanly separates: (i) the directional derivative $\mathbf{v} \cdot \nabla u$, concentrating localized (high-wavenumber) variations; and (ii) the smoothing terms, which induce global or scale-local couplings (low wavenumbers). Guided by this principle, a full-spectrum representation should preserve global spectral efficiency for low frequencies while explicitly exposing directional derivatives to complement the global description and mitigate spectral bias. The Riesz transform—realizing normalized first-order derivatives in the spectral domain—serves this purpose without committing to a specific PDE class.

**Structured rank reduction.** The global pathway performs a low-rank approximation in the Riesz basis, which is well-suited for smooth, slowly varying dynamics. In contrast, localized or anisotropic features require high rank in isotropic bases but admit a sparse, orientation-aware encoding in directional-derivative channels. RNO exploits this by replacing a large local kernel with a small directional mixerA.10: a few weights $w = (w_1, \ldots, w_d)$ acting on $\{R_i(q)\}$,

$$z \ = \ z_{\text{global}} \ + \ \sum_{i=1}^{d} w_i\, R_i(q).$$

This realizes a structured "low-rank + directional-sparse" decomposition: low-rank global modes for low frequencies, and a light, direction-sparse mixer for local high-frequency content.

**How directional derivatives are learned.** Differentiating $\mathcal{L}$ w.r.t. $w_i$ yields

$$\frac{\partial \mathcal{L}}{\partial w_i} \ = \ \Big\langle \frac{\partial \mathcal{L}}{\partial z},\, R_i(q) \Big\rangle,$$

i.e., the residual is projected onto each directional channel. Because $\{R_i(q)\}$ are approximately orthogonal, these projections decouple, and training selectively amplifies the oriented components that most reduce error. Moreover, Riesz features (A.1) are rotation-covariant: a rotation of the input induces the same rotation of the channel vector, so the learned mixing generalizes across orientations without relearning. A bounded amplitude (A.6) prevents excessive high-frequency gain, ensuring numerical stability.

## A.10 Implementation of the Direction Mixer

The direction mixer essentially performs global–local integration of physical dynamics in the Riesz space. This differs from mixing in the spatial domain, since data representations in the Riesz space are inherently distinct, and can be regarded as a latent feature representation. Consequently, the challenge is to design a fusion mechanism that preserves data characteristics while naturally integrating physical properties. In the physical domain, the monogenic signal (Unser et al., 2009), a high-dimensional extension of analytic signals, offers such an example, as it naturally fuses orthogonal components across scales. For a monogenic signal, its general form in $\mathbb{R}^3$ can be expressed as

$$f(x) = x_1 + x_2 i + x_3 j. \tag{37}$$

Our implementation follows this formulation with several modifications:

$$Mixer = R_{\text{global}} + w_i R_i + w_j R_j, \tag{38}$$

where $w$ denotes learnable, direction-specific parameters that assign spectral weights to the corresponding Riesz components. For $n$-dimensional data, the formulation naturally extends to $n$ dimensions. To avoid excessive artifacts according to the derived bounds, the values are constrained within $(0, 2)$ according to bound in A.6. This structure enables a natural fusion of global and local information, which is key to the superior performance of the overall model.

## A.11 Directionality in One Dimension

In one spatial dimension, the Riesz transform reduces to the Hilbert transform (Cizek, 1970), the canonical operator for building analytic signals (Marple, 1999):

$$\mathcal{H}\{f\}(t) = \frac{1}{\pi}\,\text{p.v.} \int_{-\infty}^{\infty} \frac{f(\tau)}{t-\tau}\,d\tau. \tag{39}$$

Table 7: Comparison of model size, training efficiency, memory footprint, and accuracy (relative $\ell_2$ error) for existing operator-learning baselines and our RNO on the Reaction-Diff. dataset.

| Model | Params (M) $\downarrow$ | Train time (s/epoch) $\downarrow$ | Peak memory consumption $\downarrow$ | Relative $\ell_2$ $\downarrow$ |
|---|---|---|---|---|
| DeepONet | 567809 | 0.263 | 246.30 MB | 0.7969 |
| FNO | 310913 | 0.098 | 230.67MB | 0.1214 |
| WNO | 192833 | 0.112 | 281.11 MB | 0.1713 |
| LNO | 64241 | 0.298 | 1635.57 MB | 0.1355 |
| RNO (ours) | 172229 | 0.052 | 111.17MB | 0.0899 |

Table 8: Comparison of model size, training efficiency, memory footprint, and accuracy (relative $\ell_2$ error) for existing operator-learning baselines and our RNO on the Diffusion dataset.

| Model | Params (M) $\downarrow$ | Train time (s/epoch) $\downarrow$ | Peak memory consumption $\downarrow$ | Relative $\ell_2$ $\downarrow$ |
|---|---|---|---|---|
| DeepONet | 403073 | 0.285 | 365.67 MB | 0.1356 |
| FNO | 527745 | 0.278 | 313.36 MB | 0.0229 |
| WNO | 192833 | 0.112 | 281.11 MB | 0.0179 |
| LNO | 76369 | 0.512 | 8805.78 MB MB | 0.0081 |
| RNO (ours) | 150609 | 0.462 | 245.71 MB | 0.0079 |

While a single spatial dimension lacks an intrinsic notion of orientation, the Hilbert transform nonetheless decomposes any real-valued signal into two oppositely directed components, distinguished by the sign of their temporal frequencies. Define

$$f_+(t) = \tfrac{1}{2}\big[f(t) + i\,\mathcal{H}\{f\}(t)\big], \tag{40}$$

$$f_-(t) = \tfrac{1}{2}\big[f(t) - i\,\mathcal{H}\{f\}(t)\big], \tag{41}$$

so that positive frequencies correspond to energy (or information) propagating away from the source, while negative frequencies correspond to energy returning toward it. The spectral imbalance between these two sectors thus reveals the net propagation direction. In fluid dynamics, spectra dominated by positive frequencies typically indicate pressure waves radiating from a point source. When solutions of the Navier–Stokes equations predict strong vortical feedback, analysis should instead focus on the negative-frequency band. In contrast, in purely diffusive processes where concentrations relax from high to low, the positive frequency component typically dominates.

## A.12 EFFICIENCY ANALYSIS

Full model efficiency results: on Reaction–Diff. (Table 7), RNO achieves the lowest error (0.0899), improving over FNO (0.1214) and LNO (0.1355) by $26\%$ and $34\%$, while being the fastest ($0.052\,\mathrm{s/epoch}$; $\sim 2\times$ vs. FNO, $> 5\times$ vs. LNO). It remains compact ($1.7\times$ fewer parameters than FNO) and memory-efficient (about half the peak memory of Fourier-based baselines and nearly $15\times$ below LNO). On Diffusion (Table 8), RNO again attains the best error (0.0079), slightly ahead of LNO (0.0081) and well ahead of FNO/WNO ($65\%/56\%$), with only a modest training-time overhead; the model is still small ($1.5 \times 10^5$ parameters, $\sim 3.5\times$ fewer than FNO) and memory usage is comparable to DeepONet and markedly below LNO. Overall, the tables indicate that RNO offers the most favorable accuracy–efficiency–memory trade-off among compared operators.

## A.13 DERIVATIVE FIDELITY.

To further evaluate RNO's ability to recover higher-order quantities, we examine the reconstruction of physically meaningful first–order derivatives on the Navier–Stokes benchmark at $\mathrm{Re} = 5000$ by comparing vorticity $\omega = \partial_x u_y - \partial_y u_x$ (middle row of Fig. 10) and divergence $\delta = \partial_x u_x + \partial_y u_y$ (bottom row). These quantities separate rotational and compressive components and are sensitive to high–wavenumber errors around shear layers and filaments.

In the vorticity panels, RNO recovers thin vortex sheets and interface polarity with high fidelity, exhibiting fewer ringing artefacts than LNO and less oversmoothing than WNO/FNO; DeepONet

Table 9: Comparison of relative $\ell_2$ errors between RNO and several high–frequency–enhanced operator baselines (WNO, LNO, loglo-FNO, and E-FNO) on the Reaction–Diffusion and Brusselator benchmarks.

| Model | Reac–Diff. | Brusselator |
|---|---|---|
| WNO | 0.1713 | 0.1789 |
| LNO | 0.1355 | 0.1858 |
| loglo-FNO | 0.1009 | 0.1679 |
| E-FNO | 0.1131 | 0.1703 |
| **RNO (ours)** | **0.0899** | **0.1317** |

largely misses fine eddies. In the divergence panels, RNO produces near-zero residuals—consistent with incompressibility—whereas the baselines show larger coherent divergence patches, especially near strongly curved structures.

The qualitative ranking in Fig. 10 indicates that the proposed directional–spectral representation improves derivative recovery in practice, capturing rotational structure while suppressing non-physical divergence.

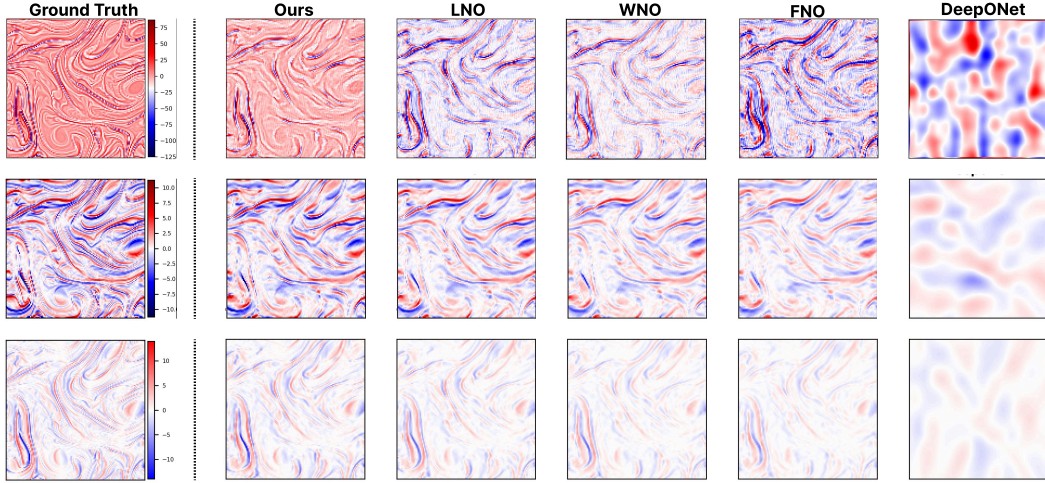

Figure 10: Comparison of high-frequency components(top), vorticity(middle), divergence(bottom) with the Ground Truth.

### A.14 HIGH-FREQUENCY BASELINES.

To further demonstrate the advantages of our method over other approaches that enhance local high-frequency components, we added a high-frequency baseline comparison on the Reaction–Diffusion and Brusselator benchmarks. The baselines include LNO, WNO, E-FNO, and loglo-FNO. Table 9 summarizes the relative $\ell_2$ errors on the Reaction–Diffusion and Brusselator benchmarks. Our RNO consistently achieves the lowest error on both datasets: it improves over the best baseline (loglo-FNO) from 0.1009 to 0.0899 on Reaction–Diffusion and from 0.1679 to 0.1317 on Brusselator. Other variants such as WNO, LNO, and E-FNO are clearly less accurate, indicating that the proposed RNO architecture transfers better across different nonlinear reaction–diffusion systems.

### A.15 ROBUSTNESS TO NOISY INPUTS.

To test whether the spectral derivative in RNO amplifies high-frequency noise, we inject additive zero-mean Gaussian noise at relative levels $\alpha \in \{0, 0.05, 0.1, 0.2\}$ into the Reaction–Diffusion inputs at test time (Table 10; $\alpha$ is the standard deviation normalized by the input scale).

Table 10: Relative $\ell_2$ error on the Reaction–Diffusion dataset under different noise levels (SNR).

| Model | 0 SNR | 0.05 SNR | 0.1 SNR | 0.2 SNR |
|-------|-------|----------|---------|---------|
| FNO | 0.1214 | 0.2129 | 0.3137 | 0.4976 |
| LNO | 0.1355 | 0.2632 | 0.3719 | 0.6622 |
| RNO | 0.0899 | 0.0926 | 0.0934 | 0.0958 |

RNO attains the lowest relative $\ell_2$ error for all $\alpha$ and degrades only mildly: $0.0899 \rightarrow 0.0958$ ($< 7\%$ increase). In contrast, FNO and LNO grow substantially with noise, from $0.1214 \rightarrow 0.4976$ and $0.1355 \rightarrow 0.6622$, respectively (roughly $+75\%$–$310\%$ and $+94\%$–$389\%$ across $\alpha$).

The stability aligns with RNO's directional-derivative representation: Riesz multipliers $m_i(\xi) = -\mathrm{i}\,\xi_i/(\|\xi\|)$ satisfy $\sum_i |m_i(\xi)|^2 = 1$ (energy-preserving), avoiding uncontrolled high-wavenumber gain, while the learned mixer weights gate directional contributions. This yields fine-scale fidelity without sacrificing robustness to realistic measurement noise.

## A.16 IMPLEMENTAION DETAILS.

All experiments are repeated three times, implemented in PyTorch and conducted on a single NVIDIA RTX 3090 GPU (two RTX 3090 GPUs were used for the ERA5 experiments.). For all methods, the performance at the final epoch is recorded as the final result. For fairness in comparison, we re-ran all baseline methods under a unified configuration at the same benchmarks.On each dataset, we use the same learning rate and number of training epochs across all methods; all models are optimized with Adam. For architectures that follow the classical neural-operator design, we align the number of modes and the model width (see Table 11). U-Net–style baselines, such as U-Net and LSM, are implemented with a symmetric 6-layer encoder–decoder structure. The implementation details of RNO on the various PDE benchmarks are summarized in Table 11.

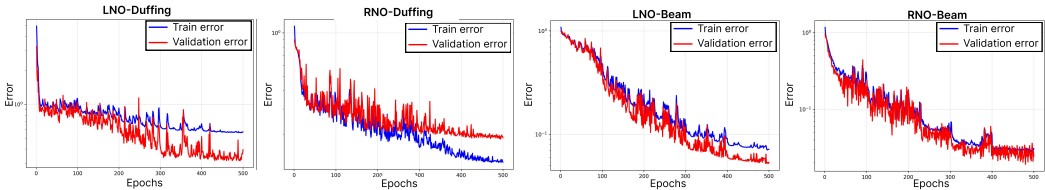

Figure 11: Training loss of LNO and RNO on Duffing and Beam datasets.

It is noted that this unified configuration leads to mild discrepancies relative to the settings used in the original papers. For example, LNO is trained for 1000 epochs on Duffing in the original work, whereas we use 500 epochs here; on Beam, we adopt different choices of modes and width. These differences explain the variation in the reported numbers. For completeness, Figure 11 shows the training loss curves of LNO and RNO on the Duffing and Beam datasets under our setup.

### A.16.1 MODEL ARCHITECTURE

**Algorithmic outline: Riesz Neural Operator (RNO).** Given $f : \Omega_h \to \mathbb{R}^{C_{\mathrm{in}}}$ on a grid $\Omega_h \subset \mathbb{R}^d$, we construct lifted features $h^{(0)} : \Omega_h \to \mathbb{R}^C$ via a pointwise map $\phi_{\mathrm{init}}$ that can depend on both $f(x)$

Table 11: RNO hyperparameters on different PDE benchmarks.

| Dataset | Batch size | Epochs | Learning rate | Width | Modes |
|---|---|---|---|---|---|
| Duffing | 20 | 500 | 0.002 | 4 | 16 |
| Beam | 40 | 500 | 0.002 | 4 | 16, 16 |
| Diffusion | 20 | 500 | 0.002 | 16 | 16, 16 |
| Reaction–Diff. | 40 | 300 | 0.002 | 36 | 8, 8 |
| Brusselator | 40 | 300 | 0.005 | 8 | 8, 8, 8 |

and the coordinates $x$. Each of the $L$ spectral layers (in the implementation $L = 1$) performs

$$\widehat{H}^{(\ell)}(\xi) = \mathcal{F}_d\big[h^{(\ell)}\big](\xi), \tag{42}$$

$$R_i(\xi) = \begin{cases} i\,\dfrac{\xi_i}{\|\xi\|_2}, & \|\xi\|_2 > 0, \\ 0, & \text{otherwise}, \end{cases} \qquad i = 1, \dots, d, \tag{43}$$

$$\widehat{R}^{(\ell)}(\xi) = \alpha_0^{(\ell)}\widehat{H}^{(\ell)}(\xi) + \sum_{i=1}^{d}\alpha_i^{(\ell)}R_i(\xi)\,\widehat{H}^{(\ell)}(\xi), \tag{44}$$

$$\widehat{Z}^{(\ell)}(\xi) = W^{(\ell)}(\xi)\,\widehat{R}^{(\ell)}(\xi), \qquad |\xi_j| \le m_j,\ j = 1, \dots, d, \tag{45}$$

$$z^{(\ell)} = \mathrm{Norm}\Big(\mathcal{F}_d^{-1}\big[\widehat{Z}^{(\ell)}\big]\Big), \tag{46}$$

$$w^{(\ell)}(x) = \phi_{\mathrm{loc}}^{(\ell)}\big(h^{(\ell)}(x)\big), \tag{47}$$

$$h^{(\ell+1)}(x) = z^{(\ell)}(x) + w^{(\ell)}(x). \tag{48}$$

Here $R_i$ are Riesz multipliers implementing normalized directional derivatives; $\alpha_0^{(\ell)}, \dots, \alpha_d^{(\ell)}$ are learnable scalar scale parameters (in the 2D implementation, $\alpha_0, \alpha_1, \alpha_2$ correspond to `scale_x`, `scale_r1`, `scale_r2`); $W^{(\ell)}(\xi) \in \mathbb{C}^{C \times C}$ is a complex-valued Riesz multiplier supported; $\phi_{\mathrm{loc}}^{(\ell)}$ is a $1 \times \cdots \times 1$ convolution acting pointwise. A normalization layer (e.g. instance normalization) is denoted by $\mathrm{Norm}$.

After $L$ layers, a pointwise readout MLP produces the prediction

$$r(x) = \sigma\big(W_1 h^{(L)}(x)\big), \qquad \sigma(t) = \sin t, \tag{49}$$

$$u_{\mathrm{pred}}(x) = W_2 r(x), \tag{50}$$

which matches the `fc1–sin–fc2` block in the implementation.

**Algorithmic outline: Spatial–Riesz Neural Operator (S-RNO).** Algorithm 2 replaces the Fourier-domain Riesz step with a filter-based approximation. For $f : \Omega_h \times \mathcal{S} \to \mathbb{R}^{C_{\mathrm{in}}}$, each layer convolves current features with a bank $\{a_i\}_{i=1}^{d}$ of fixed FIR filters whose frequency responses approximate $-\mathrm{i}\,\xi_i/\|\xi\|$, producing directional Riesz components $r_i = a_i * h^{(\ell)}$. These define a monogenic/quaternionic coefficient from which we extract local phase and amplitude along the distinguished axis $\mathcal{S}$; optional band-pass and amplitude-weighted smoothing stabilize the estimate. A scalar phase gain then rotates the Riesz coefficient, and the real part yields a phase-modulated feature that is mixed across channels and fused with a local residual branch via the nonlinearity, followed by a pointwise readout to $u_{\mathrm{pred}}$. In short, S-RNO keeps Riesz-style orientation sensitivity but performs it with compact spatial filters and explicit phase handling.

**Complexity comparison.** Let $N = |\Omega_h|$ be the number of grid points and $C_{\mathrm{in/out}}$ the channel counts. Per layer, RNO (spectral) incurs

$$\underbrace{\mathcal{O}\big(N \log N \cdot C_{\mathrm{in}}\big)}_{\text{RT/IRT}} + \underbrace{\mathcal{O}\big(d\,N\,C_{\mathrm{in}}\big)}_{\text{Riesz multipliers}} + \underbrace{\mathcal{O}\big(N\,C_{\mathrm{in}}C_{\mathrm{out}}\big)}_{\text{pointwise mixer}},$$

with memory $\Theta(N\,C)$ and overhead linear in $d$ (grid-agnostic; no spatial $k^2$ factor). S-RNO (filter-based) avoids FFTs but applies $d$ small kernels per channel:

$$\underbrace{\mathcal{O}\!\left(d\,k^d\,N\,C_{\text{in}}\right)}_{\text{FIR Riesz approx.}} + \underbrace{\mathcal{O}\!\left(N\,C_{\text{in}}C_{\text{out}}\right)}_{\text{channel mixing}},$$

with memory also $\Theta(N\,C)$. Hence S-RNO is attractive on small grids or very small $k$, whereas for high resolutions (large $N$) the RNO is typically cheaper when $k^d \gtrsim \log N$. Both remain dimension-agnostic; RNO's cost scales with $d$ via elementwise multipliers, and S-RNO via $d$ filter applications. Therefore, we adopt a frequency-domain implementation while also using frequency-domain caching to accelerate computation.

## A.17    EVALUATION METRICS

We evaluate model performance using two standard metrics in our work: the relative $\ell_2$ error and the mean squared error (MSE). The relative $\ell_2$ error measures the discrepancy normalized by the ground-truth energy and is defined as

$$\text{relative } \ell_2 \text{ error} = \frac{\sum_{i=1}^{M}\left(Y_i - \hat{Y}_i\right)^2}{\sum_{i=1}^{M}\left(Y_i\right)^2} = \frac{\|Y - \hat{Y}\|_2^2}{\|Y\|_2^2}, \tag{51}$$

The MSE quantifies the average squared prediction error:

$$\text{MSE} = \frac{1}{M}\sum_{i=1}^{M}\left(Y_i - \hat{Y}_i\right)^2. \tag{52}$$

Here $Y_i$ denotes the ground-truth value, $\hat{Y}_i$ the model prediction, and $M$ the number of samples/points.

For a (real-valued) signal $u(x)$ with Fourier transform $\hat{u}(k) = \mathcal{F}\{u\}(k)$, we define its power spectral density (PSD) as

$$S_u(k) = |\hat{u}(k)|^2, \tag{53}$$

which describes how the signal power (variance) is distributed over frequencies $k$.

## A.18    SUPPLEMENTARY FOR BENCHMARKS

### A.18.1    DUFFING OSCILLATOR

A paradigmatic non-linear, damped, and driven oscillator, the Duffing system is governed by

$$m\ddot{x}(t) + c\dot{x}(t) + k_1 x(t) + k_3 x^3(t) = f(t), \tag{54}$$

where $x(t)$ is the displacement, $\dot{x}(t)$ and $\ddot{x}(t)$ denote velocity and acceleration, and the constants $m$, $c$, $k_1$, and $k_3$ represent the mass, viscous damping, linear stiffness, and cubic stiffness, respectively. The term $f(t)$ supplies the external forcing.

### A.18.2    EULER–BERNOULLI BEAM

For a slender Euler–Bernoulli beam subject to transverse loading, the Euler–Lagrange formalism yields

$$EI\,\frac{\partial^4 y(x,t)}{\partial x^4} + \rho A\,\frac{\partial^2 y(x,t)}{\partial t^2} = f(x,t), \tag{55}$$

with $y(x,t)$ denoting the transverse deflection and $f(x,t)$ the applied load. Material and geometric properties enter through Young's modulus $E$, second moment of area $I$, density $\rho$, and cross-sectional area $A$.

### A.18.3    DIFFUSION EQUATION

Pure diffusion in one dimension obeys

$$D\,\frac{\partial^2 y(x,t)}{\partial x^2} - \frac{\partial y(x,t)}{\partial t} = f(x,t), \tag{56}$$

where $y(x,t)$ is the scalar field, $D$ the (constant) diffusion coefficient, and $f(x,t)$ a distributed source or sink.

---

**Algorithm 1** Riesz Neural Operator in $d$ dimensions

---

**Require:** Spatial dimension $d \in \{1, 2, 3\}$, grid $\Omega_h = \{x_n\}_{n=1}^N \subset \mathbb{R}^d$
**Require:** Input channels $C_{\text{in}}$, hidden width $C$, output channels $C_{\text{out}}$
**Require:** Number of spectral layers $L$ (in the implementation $L = 1$)
 1: **Lift (pointwise embedding):**
 2: **for** $n = 1$ **to** $N$ **do**
 3:     $h^{(0)}(x_n) \leftarrow \phi_{\text{init}}\big(f(x_n), x_n\big) \in \mathbb{R}^C$ ▷ e.g. concatenate $f(x_n)$ and coordinates $x_n$ and apply a linear layer
 4: **end for**
 5: **for** $\ell = 0$ **to** $L - 1$ **do**
 6:     **(a) Adaptive Riesz spectral update**
 7:     Reshape $h^{(\ell)}$ to $H \in \mathbb{R}^{C \times N_1 \times \cdots \times N_d}$
 8:     $\widehat{H} \leftarrow \mathcal{F}_d(H)$                                    ▷ $d$-D RT
 9:     For each frequency index $\kappa$ with wave vector $\xi(\kappa) \in \mathbb{R}^d$, define

$$R_i(\kappa) = \begin{cases} i \dfrac{\xi_i(\kappa)}{\|\xi(\kappa)\|_2}, & \|\xi(\kappa)\|_2 > 0, \\ 0, & \text{otherwise}, \end{cases} \quad i = 1, \ldots, d.$$

10:     Combine identity and Riesz channels with learnable scales

$$\widehat{R}(c, \kappa) \leftarrow \alpha_0^{(\ell)} \, \widehat{H}(c, \kappa) + \sum_{i=1}^d \alpha_i^{(\ell)} \, R_i(\kappa) \, \widehat{H}(c, \kappa),$$

       (in 2D, $\alpha_0, \alpha_1, \alpha_2$ correspond to `scale_x`, `scale_r1`, `scale_r2`).
11:     Initialize $\widehat{V}$ with zeros (same shape as $\widehat{R}$)
12:     **for all** $\kappa$ with $|\kappa_j| \leq m_j$ for $j = 1, \ldots, d$ **do**
13:        $\widehat{V}(:, \kappa) \leftarrow W_\ell(\kappa) \, \widehat{R}(:, \kappa)$        ▷ $W_\ell(\kappa) \in \mathbb{C}^{C \times C}$, implemented by complex weights
14:     **end for**
15:     $V \leftarrow \mathcal{F}_d^{-1}(\widehat{V})$                                        ▷ inverse RT
16:     $z^{(\ell)} \leftarrow \text{Norm}(V)$ ▷ e.g. instance normalization; in code you may normalize before/after the Riesz layer
17:     **(b) Local residual update and fusion**
18:     **for** $n = 1$ **to** $N$ **do**
19:        $w^{(\ell)}(x_n) \leftarrow \phi_{\text{loc}}^{(\ell)}\big(h^{(\ell)}(x_n)\big)$             ▷ $1 \times \cdots \times 1$ convolution, e.g. `Conv2d(width,width,1)`
20:        $h^{(\ell+1)}(x_n) \leftarrow z^{(\ell)}(x_n) + w^{(\ell)}(x_n)$
21:     **end for**
22: **end for**
23: **Readout (pointwise regression):**
24: **for** $n = 1$ **to** $N$ **do**
25:     $r(x_n) \leftarrow \sigma\big(W_1 h^{(L)}(x_n)\big)$       ▷ in the implementation $\sigma(t) = \sin(t)$, layer `fc1 + sin`
26:     $u_{\text{pred}}(x_n) \leftarrow W_2 r(x_n) \in \mathbb{R}^{C_{\text{out}}}$                          ▷ final linear layer `fc2`
27: **end for**
28: **return** $u_{\text{pred}}$

---

### A.18.4    REACTION–DIFFUSION SYSTEM

Coupling diffusion with non-linear kinetics leads to the generic reaction–diffusion form

$$D \frac{\partial^2 y(x, t)}{\partial x^2} + k \, y^2(x, t) - \frac{\partial y(x, t)}{\partial t} = f(x, t), \tag{57}$$

where $k$ sets the quadratic reaction rate while $y(x, t)$ and $f(x, t)$ retain their usual interpretations.

---

**Algorithm 2** Spatial-Riesz Neural Operator (PRNO) with filter-based Riesz approximation

---

**Require:** Spatial dimension $d \in \{1, 2, 3\}$, grid $\Omega_h = \{x_n\}_{n=1}^N \subset \mathbb{R}^d$
**Require:** Distinguished 1D axis $\mathcal{S} = \{1, \ldots, T\}$ (e.g., time)
**Require:** Input channels $C_{\text{in}}$, width $C$, output channels $C_{\text{out}}$
**Require:** Temporal band $(f_{\text{low}}, f_{\text{high}})$ at sampling rate $f_s$, amplification $\alpha$
**Require:** Number of layers $L$
**Require:** Spatial FIR filters $\{a_j\}_{j=1}^d$ approximating Riesz components (i.e., DTFT $D_j(\omega) \approx \omega_j/\|\omega\|_2$ on the band of interest), temporal bandpass filter $\mathcal{T}$ on $(f_{\text{low}}, f_{\text{high}})$, spatial Gaussian blur kernel $g$

1: **Input–output map:**
2: Given $f : \Omega_h \times \mathcal{S} \to \mathbb{R}^{C_{\text{in}}}$, define $\mathcal{N}_\theta : f \mapsto u_{\text{pred}}$.
3: **Lift (pointwise embedding):**
4: $h^{(0)}(x_n, s) \leftarrow \phi_{\text{lift}}(f(x_n, s)) \in \mathbb{R}^C$ for all $(x_n, s)$
5: **for** $\ell = 0 \; L - 1$ **do**
6:     **(a) Filter-based Riesz responses on $\Omega_h$:**
7:     $I^{(\ell)}(x_n, s) \leftarrow h^{(\ell)}(x_n, s)$
8:     $R_j^{(\ell)}(x_n, s) \leftarrow (a_j * I^{(\ell)}(\cdot, s))(x_n)$ for $j = 1, \ldots, d$             $\triangleright$ $d$-D conv on $\Omega_h$
9:     $q^{(\ell)}(x_n, s) \leftarrow \big(I^{(\ell)}(x_n, s), R_1^{(\ell)}(x_n, s), \ldots, R_d^{(\ell)}(x_n, s)\big)$
10:    **(b) Quaternionic phase pipeline along $\mathcal{S}$:**
11:    $q^\Delta(x_n, s) \leftarrow q^{(\ell)}(x_n, s) \, \overline{q^{(\ell)}(x_n, s-1)}$ (with $s = 1$ using $s - 1 = 1$)
12:    $(\Delta\phi(x_n, s), A(x_n, s)) \leftarrow \text{PHASEDIFFANDAMP}(q^\Delta(x_n, s))$
13:    $\phi_u(x_n, \cdot) \leftarrow$ cumulative sum of $\Delta\phi(x_n, \cdot)$ over $s$
14:    $\phi_f(x_n, \cdot) \leftarrow \mathcal{T}(\phi_u(x_n, \cdot))$                       $\triangleright$ temporal bandpass
15:    $\phi_s(\cdot, s) \leftarrow \text{AMPWEIGHTEDBLUR}(\phi_f(\cdot, s), A(\cdot, s); g)$     $\triangleright$ spatial blur on $\Omega_h$
16:    $\phi_a(x_n, s) \leftarrow \alpha \, \phi_s(x_n, s)$
17:    $q^{\text{mag}}(x_n, s) \leftarrow \text{PHASESHIFT}(q^{(\ell)}(x_n, s), \phi_a(x_n, s))$
18:    $I^{\text{mag}}(x_n, s) \leftarrow \text{REALPART}(q^{\text{mag}}(x_n, s))$
19:    **(c) Nonlocal–local fusion:**
20:    $z^{(\ell)}(x_n, s) \leftarrow \phi_{\text{mix}}^{(\ell)}(I^{\text{mag}}(x_n, s))$             $\triangleright$ $1 \times \cdots \times 1$ channel mixing
21:    $w^{(\ell)}(x_n, s) \leftarrow \phi_{\text{loc}}^{(\ell)}(h^{(\ell)}(x_n, s))$
22:    $h^{(\ell+1)}(x_n, s) \leftarrow \sigma(z^{(\ell)}(x_n, s) + w^{(\ell)}(x_n, s))$
23: **end for**
24: **Readout (pointwise regression):**
25: $u_{\text{pred}}(x_n, s) \leftarrow \phi_{\text{out}}(h^{(L)}(x_n, s)) \in \mathbb{R}^{C_{\text{out}}}$
26: **return** $u_{\text{pred}}$

---

### A.18.5 BRUSSELATOR REACTION–DIFFUSION SYSTEM

The two-species Brusselator, describing an autocatalytic chemical network, is written as

$$
\begin{aligned}
\frac{\partial u}{\partial t} &= D_0\left(\frac{\partial^2 u}{\partial x^2} + \frac{\partial^2 u}{\partial y^2}\right) + a + f(t) - (1 + b)u + vu^2, \\
\frac{\partial v}{\partial t} &= D_1\left(\frac{\partial^2 v}{\partial x^2} + \frac{\partial^2 v}{\partial y^2}\right) + bu - vu^2,
\end{aligned}
\tag{58}
$$

defined for $(x, y) \in [0, 1]^2$ and $t \in [0, 20]$. Here $u$ and $v$ are concentration fields, $D_0$ and $D_1$ their diffusion coefficients, and $a, b$ the kinetic parameters, with $f(t)$ introducing a time-dependent feed term.

### A.18.6 NAVIER–STOKES EQUATION

For an incompressible Newtonian fluid with velocity $\mathbf{u}(\mathbf{x}, t)$ and pressure $p(\mathbf{x}, t)$,

$$
\partial_t \mathbf{u} + (\mathbf{u} \cdot \nabla)\mathbf{u} = -\nabla p + \nu \Delta \mathbf{u} + \mathbf{f}, \qquad \nabla \cdot \mathbf{u} = 0,
$$

where $\nu$ denotes the kinematic viscosity and $\mathbf{f}$ a body force. Upon nondimensionalization using the characteristic speed $U$ and length $L$,

$$\partial_t \mathbf{u} + (\mathbf{u} \cdot \nabla)\mathbf{u} = -\nabla p + \frac{1}{Re}\Delta \mathbf{u} + \mathbf{f}, \qquad \nabla \cdot \mathbf{u} = 0,$$

the sole control parameter is the Reynolds number $Re = UL/\nu$, which quantifies the ratio of inertial to viscous effects. Higher Reynolds numbers are accompanied by increasingly complex local structures, which is one of the challenges faced by current neural-network methods. Accordingly, using configurations spanning a range of Reynolds numbers in this experiment to validate the model's capacity to represent local nonlinearities is both reasonable and reliable. As shown in Figure 12, RNO retains excellent predictive performance despite increasing local complexity.

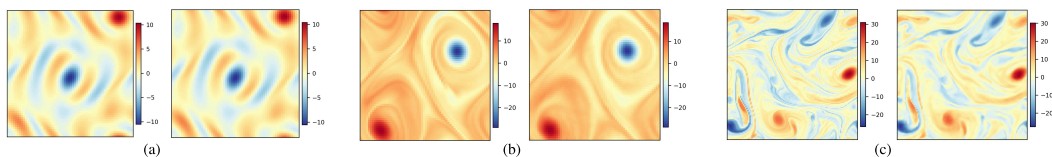

Figure 12: Comparison between the ground truth and RNO predictions across different Reynolds numbers. (a) $Re = 40$, (b) $Re = 500$, (c) $Re = 5000$.

### A.18.7 ERA5 DATASET

We use the fifth-generation ECMWF reanalysis (ERA5) as our data source and focus on the geopotential height at 500 hPa (Z500), a standard mid-tropospheric diagnostic of large-scale circulation. The native ERA5 Z500 fields are regridded to a regular $2.5°$ latitude–longitude mesh, resulting in a $72 \times 144$ global grid. We consider a continuous period from 2013 to 2023, using the years 2013–2022 for training and 2023 as a held-out test year. From this time span we construct more than 20,000 input–target sequences for our experiments. The processed dataset occupies approximately 2.5 GB on disk, of which the 2023 test split accounts for about 0.6 GB.

To reduce the influence of static geographic biases in the Z500 field and to emphasize dynamically relevant circulation anomalies, we work with grid-point anomalies rather than raw absolute values. Specifically, for each grid point $(i, j)$ and time $t$, we define

$$Z'_{i,j,t} = Z_{i,j,t} - \overline{Z}_{i,j}^{2013\text{–}2019}, \tag{59}$$

where $Z_{i,j,t}$ denotes the original Z500 value and $\overline{Z}_{i,j}^{2013\text{–}2019}$ is the multi-year climatological mean at that location computed over 2013–2019. This anomaly transformation removes the leading-order stationary spatial structure (e.g., latitudinal gradients and orographic imprints), encourages the model to focus on temporal variability and circulation patterns, and avoids information leakage because the climatology is computed without using the 2023 test year. Unless otherwise stated, all experiments and evaluation metrics are reported in terms of these anomaly fields, while absolute Z500 values can be recovered by adding back the corresponding climatology.

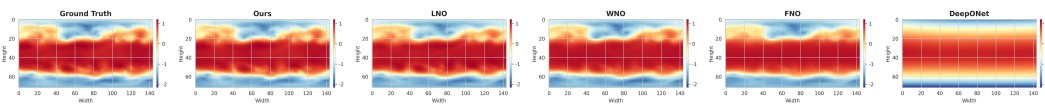

Figure 13: Solution comparison between the ground truth and predictions across ERA5.

## DISCUSSION AND LIMINATION

RNO's key insight is a scale-wise repartitioning of the data. This reallocation expands the model's representational degrees of freedom. In neural-operator architectures, it provides a principled mechanism to surface local components that would otherwise be suppressed. Although these local components may spatially overlap with global features, such overlap does not degrade performance, as

the original signals at those locations were underweighted. RNO explicitly amplifies these neglected local contributions and rebalances the component weights. From a Taylor-expansion perspective, these local terms admit further refinement, which we leave to future work.

RNO further models the coupling between the data's wave content and the neural-operator architecture, yielding a more complete justification of the design. Nonetheless, RNO has limitations: in some settings, control over local components lacks sufficient precision that is an issue we aim to address in subsequent work.

## LLM USAGE STATEMENT: LANGUAGE POLISHING ONLY

We used a large language model (LLM) solely for language polishing to improve clarity and control manuscript length (e.g., correcting grammar and refining phrasing for greater precision.). The LLM was not used for problem formulation, methodology or experiment design, data processing or analysis, result interpretation, or drawing conclusions. All edits suggested by the LLM were reviewed and approved by the authors.

