# OpenReview forum: "Riesz Neural Operator for Solving Partial Differential Equations"
_ICLR.cc/2026/Conference — ICLR 2026 Poster_

### Official Review · Reviewer_xBjE · 2025-10-28

**Soundness:** 3
**Presentation:** 3
**Contribution:** 3
**Rating:** 6
**Confidence:** 4

**Summary:**

The paper considers the problem of learning PDE solutions with neural operators, and proposes Riesz neural operator (RNO), which works on the spectral domain to improve spatial locality representation. The authors provide approximation error bounds with the use of first order Taylor expansion, and empirically show performance improvements over a couple baselines on several PDEs.

**Strengths:**

1. The idea of using Reisz transformation to capture local derivatives is interesting and seems beneficial in learning PDEs with highly changing solutions.
2. The paper is in general clearly written with definitions of the core Reisz transformation explained in the Appendix.

**Weaknesses:**

1. There seems to be a lack of comparison (both conceptually and experimentally) with an important baseline WNO [1], which uses wavelet transformation and addresses localization in both time/space and frequency.

2. A detailed training procedure is missing. A full algorithm in the appendix should help.

3. The ERA5 problem seems to serve as an important realistic application of the method, but misses key details such as training/testing data size, solution visualization etc. Providing a public link does not make the paper self consistent.

[1] Tripura, Tapas, and Souvik Chakraborty. "Wavelet neural operator: a neural operator for parametric partial differential equations." arXiv preprint arXiv:2205.02191 (2022).

**Questions:**

1. PSD is used without definition, is that power spectral density?
2. I believe the Reisz space in the paper means the Reisz field or Reisz representation in signal processing, instead of the vector lattice in functional analysis. The authors should clarify and be more accurate in wording.
3. How are the directional derivatives effectively learned during training?

---

> ### Author Response · Authors · 2025-11-24
>
> Thank you very much for your valuable suggestions and for your thoughtful appreciation of our work. I address your questions point by point below.
>
> ---
> >**W1**:
> >There seems to be a lack of comparison (both conceptually and experimentally) with an important baseline WNO [1], which uses wavelet transformation and addresses localization in both time/space and frequency.
>
> **A1**:
> Thank you for pointing this out. **WNO**[1] is indeed a fundamental baseline within this family of methods, and we agree that it should be included as one of our primary comparisons. We have added WNO results to **all** main baseline tables as well as to the newly introduced high-frequency comparison table. We also incorporated WNO into the **all** visualization results presented in the revision.
>
> Table1: Results on diverse benchmarks.
> | Model | Duffing | Beam | Diffusion | Reaction–Diff. | Brusselator |
> |-------|---------|------|-----------|----------------|-------------|
> | WNO   | 0.2532  | 0.0571 | 0.0179 | 0.1713 | 0.1789 |
>
>
> Table2: Results on NS dataset.
> | Datasets        | DeepONet | FNO     | F-FNO  | WNO    | LNO    | RNO |
> |----------------|----------|---------|--------|--------|--------|---------|
> | Re = 40        | 0.0280   | 0.0078  | 0.0065 | 0.0077 | 0.0060 | 0.0049 |
> | Re = 500       | 3.4082   | 1.4251  | 1.2312 | 0.9388 | 1.2117 | 0.4861 |
> | Re = 5000      | 6.2721   | 2.9314  | 2.8912 | 2.5914 | 2.3139 | 0.9121 |
>
>
> Table3: Results on ERA5 dataset.
> | Dataset | DeepONet | FNO    | WNO    | LNO    | RNO |
> |---------|----------|--------|--------|--------|---------|
> | ERA5    | 0.0912   | 0.0093 | 0.0085 | 0.0062 | 0.0022 |
>
> [1] Wavelet neural operator: a neural operator for parametric partial differential equations. 2022arXiv.
>
> ---
> >**W2**:
> >A detailed training procedure is missing. A full algorithm in the appendix should help.
>
> **A2**:
> We included the full pseudocode for the RNO pipeline in the added A.16, showing both the spatial-domain and frequency-domain implementations. These additions describe the implementation steps in detail and provide an analysis of the computational complexity and practical performance of the two variants.
>
> ---
> >**W3**:
> >The ERA5 problem seems to serve as an important realistic application of the method, but misses key details such as training/testing data size, solution visualization etc. Providing a public link does not make the paper self consistent.
>
> **A3**:
> We added further details on ERA5 in the A.18 to provide a more complete description of the dataset beyond the brief overview .
>
> ---
> >**Q1**:
> >PSD is used without definition, is that power spectral density?
>
> **A1**:
> Yes, PSD refers to the power spectral density. We have added a precise definition in the metrics subsection (A.17) and clarified the notation in the main text.
>
> ---
> >**Q2**:
> >I believe the Reisz space in the paper means the Reisz field or Reisz representation in signal processing, instead of the vector lattice in functional analysis. The authors should clarify and be more accurate in wording.
>
> **A2**：
> This could indeed cause confusion; thank you for the suggestion. We have added clarifications in the section of the main text(line155-158) that introduces the Riesz method.
>
>
> ---
> >**Q3**:
> >How are the directional derivatives effectively learned during training?
>
> **A3**：
> The directional derivatives are implemented in the frequency domain by extracting components along **different orientations**. This allows us to preserve the **continuity-learning** properties of neural operators while aligning with the natural signal representation given by monogenic signals, a higher-order analogue of analytic signals with a multi-dimensional complex structure. We propose an adaptive local-to-global refinement mechanism, in which the scale of each directional derivative is adjusted **adaptively** to compensate for the loss of fine details during training. This design enables more effective fusion of non-redundant components. We provide a more detailed discussion in A.9.
>
> We also thank you for the careful reading of our paper and for providing many valuable detailed comments and corrections.Your constructive suggestions have been very helpful in improving our paper.

---

### Official Review · Reviewer_Pgop · 2025-10-31

**Soundness:** 2
**Presentation:** 2
**Contribution:** 2
**Rating:** 4
**Confidence:** 3

**Summary:**

The key idea of this paper is to embed the Riesz transform, a spectral tool representing directional derivatives, into the operator learning framework. By doing so, RNO blends global spectral features with local directional variations, addressing a noted gap in existing operator-learning methods. The paper demonstrates that RNO achieves state-of-the-art accuracy on a broad suite of PDE benchmarks and a real-world dataset.

**Strengths:**

1. The use of the Riesz transform to incorporate local spatial derivatives into a neural operator is a fresh and well-motivated idea.
2. The RNO delivers consistent and significant improvements across a wide range of PDE problems.
3. The paper is supported with rigorous theoretical justification. (Though the reviewer does not have enough time to justify its correctness)

**Weaknesses:**

1. I think some of the baselines are higher than those reported in the original paper. For instance, LNO on beam & Duffing. On some datasets, this would affect the ranking of different methods. The reviewer would like some explanation on this.
2. The experiment on Navier-Stokes is not rigorous enough, given the limited baselines.
3. The writing of this paper could be hard to follow. For instance, some of the hypotheses proposed in the introduction are not fully justified later.
4. Baseline, specializing in capturing high-frequency features, is also missing.
5. The implementation code of the paper is also missing.

**Questions:**

See cons

---

> ### Author Response · Authors · 2025-11-24
>
> We sincerely thank the reviewer for their positive assessment of our theoretical analysis and constructive advice. We will explain your concern as follows.
>
> ---
> >**Q1**:
> >I think some of the baselines are higher than those reported in the original paper. For instance, LNO on beam & Duffing. On some datasets, this would affect the ranking of different methods. The reviewer would like some explanation on this.
>
> **A1**:
> Thank you for pointing this out. For fairness in comparison, we re-ran all baseline methods under a unified configuration, which explains why some numbers differ from those reported in the original papers (for example, the epoch of Duffing in the original paper is 1000, we now use 500 for fairness). We have added the detailed experimental settings—including the RNO hyperparameters and the alignment criteria used for comparable methods—in A.16, and provided explanations for the discrepancies as well as the Beam and Duffing comparison loss curves shown in Fig. 11.
>
>
>
>
> ---
> >**Q2**:
> >The experiment on Navier-Stokes is not rigorous enough, given the limited baselines.
>
> **A2**:
>
> We have expanded the Navier–Stokes experiment section by adding additional baselines, including WNO[1]—which also incorporates a notion of locality—and F-FNO. We further enriched the visualization results with side-by-side comparisons. In addition, we computed and included visualizations of vorticity (∇×u) and divergence (∇·u)to illustrate how RNO recovers first-order physical quantities in line with our theoretical motivation in Fig.10. Across all these evaluations, RNO continues to achieve the best performance.
>
> Table1: Results on NS dataset.
> | Datasets        | DeepONet | FNO     | F-FNO  | WNO    | LNO    | RNO |
> |----------------|----------|---------|--------|--------|--------|---------|
> | Re = 40        | 0.0280   | 0.0078  | 0.0065 | 0.0077 | 0.0060 | 0.0049 |
> | Re = 500       | 3.4082   | 1.4251  | 1.2312 | 0.9388 | 1.2117 | 0.4861 |
> | Re = 5000      | 6.2721   | 2.9314  | 2.8912 | 2.5914 | 2.3139 | 0.9121 |
>
> [1] Wavelet neural operator: a neural operator for parametric partial differential equations. 2022arXiv.
>
>
> ---
> >**Q3**:
> The writing of this paper could be hard to follow. For instance, some of the hypotheses proposed in the introduction are not fully justified later.
>
> **A3**:
> Thank you for the valuable suggestions regarding our theoretical discussion. We agree that additional clarification is needed to better support the claims made in the introduction, and we have updated the revision accordingly.
>
> * 3.1 First, regarding the hypothesis *“once spatial locality is captured adequately, non-stationary temporal signals can be learned naturally”*: for a local PDE of the form  ∂ₜu = F(x, t, Ψ(u)), once the spatial locality Ψ(u) (or a stable surrogate such as (u, Ru)) is accurately captured, the temporal derivative ∂ₜu becomes a continuous pointwise function of (x, t, Ψ(u)). A small network can then approximate and integrate this function, so non-stationary temporal signals are naturally learnable. We have added a detailed proof of this claim in A.2.
>
> * 3.2 Second, for the statement *“neglecting locality therefore suppresses high-frequency representations and weakens the modeling of strong nonlinearities”*: from a signal-processing perspective, high-frequency components correspond to local fine-scale features [1, 2]. In images, for example, local details correspond to high frequencies in the spectral domain, whereas the coarse global structure is captured by low-frequency components [3]. Ignoring locality therefore inevitably leads to a loss of high-frequency detail. To clarify this point, we have expanded the corresponding discussion in the introduction (lines 52–53).
>
> [1]High-frequency flow field super-resolution via physics-informed hierarchical adaptive Fourier feature networks, 2025 Physics of Fluids.
>
> [2]Of aldapfive signal processing, 1985.
>
> [3]High-frequency component helps explain the generalization of convolutional neural networks, 2020CVPR.

---

> ### Author Response · Authors · 2025-11-24
>
> ---
> >**Q4**:
> Baseline, specializing in capturing high-frequency features, is also missing.
>
> **A4**:
> We agree that it is important to compare RNO against baselines that are explicitly designed for high-frequency modeling. In the revised version, we add LNO, WNO[1] and two FNO+LocalConv variants[2][3] as strong high-frequency baselines. LNO and WNO capture high-frequency behavior by focusing on instantaneous signal characteristics, while FNO+localConv relies on external structures to extract high-frequency details.
>
> We report their performance on  Reaction–Diff. and Brusselator benchmarks. As shown in the Table, RNO consistently outperforms these high-frequency baselines in relative $\ell_2$.
>
> Table2: Relative ℓ₂ results on Reaction-Diff. and Brusselator dataset.
> | Model      | Reac–Diff. | Brusselator |
> |------------|------------|-------------|
> | WNO        | 0.1713     | 0.1789      |
> | LNO        | 0.1355     | 0.1858      |
> | loglo-FNO  | 0.1009     | 0.1679      |
> | E-FNO    | 0.1131     | 0.1703      |
> | RNO (ours) | 0.0899 | 0.1317 |
>
> [1] Wavelet neural operator: a neural operator for parametric partial differential equations. 2022arXiv.
> [2] Enhancing Fourier Neural Operators with Local Spatial Features, arxiv2025.
> [3] LOGLO-FNO: Efficient Learning of Local and Global Features in Fourier Neural Operators, arxiv2025.
>
>
> ---
> >**Q5**:
> The implementation code of the paper is also missing.
>
> **A5**:
> We have submitted our code in the supplementary material, which contains the complete implementation workflow. A cleaned and fully organized version will be moved to a public repository upon publication.
>
> If you have any additional concerns, please do not hesitate to let us know. We are more than willing to address them and sincerely appreciate your valuable feedback and support.

---

### Official Review · Reviewer_F9gf · 2025-10-31

**Soundness:** 2
**Presentation:** 3
**Contribution:** 2
**Rating:** 6
**Confidence:** 3

**Summary:**

This paper proposes the Riesz Neural Operator (RNO), which leverages the Riesz transform to enhance neural operator learning for PDEs. By mapping inputs into a Riesz domain that encodes directional and derivative information, RNO integrates both local and global spectral components to model anisotropic and high-frequency behaviors more effectively, introducing direction-aware mixing for richer feature representations. The method is supported by comprehensive experimental validation across diverse PDE settings.

**Strengths:**

This paper presents a novel integration of the Riesz transform to encode directional and derivative information within a neural operator framework. The design is theoretically grounded and suggests the potential for constructing neural operators that effectively capture both local and global information. The proposed architecture is sound and successfully extends the representational capacity of neural operators, and the paper is overall well written. Furthermore, it provides thorough experimental validation across diverse datasets and baselines. In particular, the studies on NS equations with varying Reynolds numbers and the ablation analyses of the proposed architecture were especially effective in verifying the model’s components and advantages.

**Weaknesses:**

* The paper emphasizes that using the Riesz transform enables efficient representation of higher-order derivatives. However, employing the Riesz transform is primarily a design strategy that maps inputs into a space where higher-order derivatives are representable—it does not inherently guarantee that the model will learn or reproduce higher-order derivatives more accurately. To substantiate this claim, it would be important to experimentally evaluate how well the model reconstructs derivatives of various orders (e.g., $L^2$ and $L^\infty$ errors of $\nabla u$, $\nabla^2 u$, $\nabla^3 u$).

* The modular structure and the use of complex-valued or monogenic representations may introduce nontrivial computational and memory overhead. A fair comparison of training time, inference time, peak memory allocation during training, and parameter efficiency  against baselines is needed.

* The proposed model may introduce potential instability in practical scenarios, as Riesz transforms can amplify high-frequency noise and encounter numerical issues near singularities. Therefore, experimental validation of stability under such conditions (i.e., non-smooth solutions and noisy data) is necessary. This is especially important because real-world data often contain noise, and many practical PDEs exhibit singularities; thus, these experiments are crucial to assess the practical robustness of the proposed method.

* Tables 3 and 4 could be better positioned; their current placement does not align well with the corresponding discussion in the text.

**Questions:**

Please address the noted weaknesses, particularly by providing experimental evaluations of computational complexity and memory usage compared to baselines. Additionally, testing the model’s stability and performance on noisy data and singular solutions would greatly strengthen the validation of the proposed method’s practical robustness.

---

> ### Author Response · Authors · 2025-11-24
>
> Overall, we are encouraged by the recognition and deeply appreciate your valuable suggestions. Moving forward, we will address each of your concerns point by point.
>
> ---
> >**Q1** ：
> >The paper emphasizes that using the Riesz transform enables efficient representation of higher-order derivatives. However, employing the Riesz transform is primarily a design strategy that maps inputs into a space where higher-order derivatives are representable—it does not inherently guarantee that the model will learn or reproduce higher-order derivatives more accurately. To substantiate this claim, it would be important to experimentally evaluate how well the model reconstructs derivatives of various orders (e.g., $L^2$ and $L^\infty$ errors of $\nabla u$, $\nabla^2 u$, $\nabla^3 u$).
>
> **A1**:
>
> * 1.1 First, we agree that the issue of higher-order characterization is important and valuable. However, we would like to clarify that RNO does not embed high-order representations in the network. Instead, it employs a **first-order approximation** of the Taylor expansion. The derivative terms in RNO capture small variations in the underlying PDE. When the perturbation γ(t) in Eq.(2) is small, the second- and higher-order terms become negligible relative to the first-order term. We provide a supporting derivation in A.2 to substantiate this point.
>
> * 1.2 For this reason, we also analyzed and derived a theoretical bound for this approximation in the A.6, showing that the bound depends on the wavelength of the data. Based on this analysis and empirical observations, we additionally constrain the scale of the derivative features to avoid overlapping effects.
>
> * 1.3 However, your suggestion to evaluate quantities such as $$\nabla u$$ is very helpful for further illustrating the strengths of our method. Accordingly, in the Navier–Stokes experiment at Re = 5000, we added derivative-accuracy evaluations, reporting the **L² errors of ∇u and Δu**, and, for 2D NS, vorticity and divergence. RNO shows consistent improvements over all baselines. The corresponding visual comparisons have been added to the main text and are presented as the new Fig.10. We also added a section A.13 to further discuss this issue.
>
> ---
> >**Q2**：
> >A fair comparison of training time, inference time, peak memory allocation during training, and parameter efficiency against baselines is needed.
>
> **A2**:
>
> We thank you for pointing out this omission in our original submission. In the revision, we report a comparison of parameter size, training time, and peak memory for RNO and several baselines on two different benchmarks. The results in Tables show that RNO maintains high accuracy while also being highly efficient. The corresponding analyses have been added to the main text (Sec. 12, A.16, Fig.6).
>
>
> The monogenic representation introduces minimal additional cost, as it is a lightweight mixing scheme. The real and imaginary components are combined through simple mixing (as presented in A.10 ), and all complex-valued computations still follow the standard real–imaginary formulation. The detailed explanation can be found in A16.
>
> Table1: Results on Reaction-Diff. benchmark.
> | Model        | Params (M)  | Train time (s/epoch)  | Peak memory  | Relative ℓ₂  |
> |--------------|--------------|-------------------------|----------------|----------------|
> | DeepONet     | 567,809      | 0.263                   | 246.30 MB      | 0.7969         |
> | FNO          | 310,913      | 0.098                   | 230.67 MB      | 0.1214         |
> | WNO          | 192,833      | 0.112                   | 281.11 MB      | 0.1713         |
> | LNO          | 64,241       | 0.298                   | 1,635.57 MB    | 0.1355         |
> | RNO (ours) | 172,229 | 0.052               | 111.17 MB  | 0.0899     |
>
> Table2: Results on Diffusion benchmark.
> | Model        | Params (M)  | Train time (s/epoch)  | Peak memory  | Relative ℓ₂  |
> |--------------|--------------|-------------------------|----------------|----------------|
> | DeepONet     |  403073     | 0.285                   |  365.67 MB      | 0.1356         |
> | FNO          | 527,745      | 0.278                   | 313.36 MB      | 0.0229         |
> | WNO          | 192,833      | 0.112                   | 281.11 MB      | 0.0179         |
> | LNO          | 76,369       | 0.512                   | 8,805.78 MB    | 0.0081         |
> | RNO (ours) | 150,609 | 0.462               | 245.71 MB  | 0.0079

---

> ### Author Response · Authors · 2025-11-24
>
> ---
> >**Q3**:
> >The proposed model may introduce potential instability in practical scenarios, as Riesz transforms can amplify high-frequency noise and encounter numerical issues near singularities. Therefore, experimental validation of stability under such conditions (i.e., non-smooth solutions and noisy data) is necessary. This is especially important because real-world data often contain noise, and many practical PDEs exhibit singularities; thus, these experiments are crucial to assess the practical robustness of the proposed method.
>
> **A3**:
> From a theoretical perspective, the Riesz transform is L²-bounded, with the magnitude multiplier given in Eq. (9) as
> **j·kᵢ / ‖k‖** which does not amplify high-frequency magnitudes and instead emphasizes phase and orientation (A.1). In practice, we further **bound** the directional scaling using ζ (Eq. (11); A.6) to prevent over-sharpening. RNO’s results on the real-world ERA5 dataset also confirm this.
>
> Experimentally, we added robustness tests on noisy inputs in the revision. On the Reaction–Diffusion dataset, across all SNR levels from no noise to high noise, the relative ℓ₂ error of RNO behaves similarly to LNO and FNO, demonstrating strong robustness to noise. We have added the detailed analysis to the paper as A.15.
>
>
> | Model | 0 SNR  | 0.05 SNR | 0.1 SNR | 0.2 SNR |
> |-------|--------|----------|---------|---------|
> | FNO   | 0.1214 | 0.2129   | 0.3137  | 0.4976  |
> | LNO   | 0.1355 | 0.2632   | 0.3719  | 0.6622  |
> | RNO   | 0.0899 | 0.0926   | 0.0934  | 0.0958  |
>
> ---
> >**Q4**:
> >Tables 3 and 4 could be better positioned; their current placement does not align well with the corresponding discussion in the text.
>
> **A4**:
> We have updated the revised version accordingly to improve the readability of the paper.
>
> Your feedback is highly appreciated and has helped us further refine and strengthen the paper.

---

### Official Review · Reviewer_UKYE · 2025-11-01

**Soundness:** 2
**Presentation:** 3
**Contribution:** 3
**Rating:** 4
**Confidence:** 4

**Summary:**

The paper proposes the Riesz Neural Operator, a novel architecture designed to enhance the ability of neural operators to model complex dynamic variations. The key insight is to overcome the limitation of current methods that often overlook or collapse local non-stationarity and spatial locality. The authors demonstrate that RNO achieves better prediction accuracy and generalization performance on various PDE benchmarks, including Navier-Stokes and the real-world ERA5 climate dataset.

**Strengths:**

- The central idea of explicitly embedding the Riesz transform to extract direction-aware spectral derivatives.

- Results on 2D NS and the ERA5 climate dataset shows improvement on high-frequency modeling.

**Weaknesses:**

- Lack of complexity analysis: This new neural operator architecture needs be accompanied by an analysis of its computational overhead, especially given that Riesz transform is not a popular thing in terms of both a analysis tool and practical implementations. The paper didn't provide comparison of training time, memory consumption, or parameter count against popular baselines, which are known to have very efficient implementations. If RNO is implemented in terms of FFT, it becomes crucial to ablate the model design (see below).

- Novelty of local features. The local-global split has already been applied to standard FNOs using e.g. local convolutions. Given the relation between Riesz transform and FFT, it is not clear whether the improvement comes from the representation itself, or a carefully tuned model architecture/hyper-parameter.

- Motivation of derivatives. The authors motivate their architecture by emphasizing the importance of derivatives in solving PDEs. While this is a generally safe statement, the characteristics of different PDE classes makes it less convincing. There are many (spatially) localized PDEs, but there are also PDEs that are global, or even scale-local. Motivating an architecture based on a vague view of all PDEs does not convince me. Nevertheless, I acknowledge that spectral bias is a very significant deficiency of neural operators. I suggest the authors to emphasize on how RNO properly models the full spectrum, why different rank-reduction method are useful for global vs local dynamics.

**Questions:**

- What is the complexity of your Riesz transform implementation.

- Can you differentiate your method from FNO + CNN?

---

> ### Author Response · Authors · 2025-11-24
>
> We sincerely appreciate your constructive and insightful comments. We will explain your concerns point by point.
>
> ---
> > **Q1:**：
> > What is the complexity of your Riesz transform implementation.
>
> **A1**:
> A1:
> We acknowledge that this aspect was missing from our experimental evaluation and and further extend it.
>
> * 1.1 Theoretical complexity.
> We implemented and analyzed two variants:
> (i) a spatial-domain FIR-based Riesz transform (S-RNO) following [1,2], and
> (ii) a FFT-based spectral variant.
> For an input of size N with a k×k FIR kernel, the spatial version has O(N k²) complexity, while the frequency-based version costs O(N log N). Meanwhile, we cache the directional frequency components to reduce computational overhead.
>
> * 2.1 Practical efficiency.
> In practice, the spatial FIR implementation is about 2.5× slower and slightly less accurate under the same settings, so our final RNO uses the FFT-based variant. We now make this explicit in App. A.16, where we provide pseudocode and a complexity comparison. In addition, Sec. 4.2 and Fig. 6 report end-to-end training time, parameter counts, and peak memory on the Diffusion and Reaction–Diff. datasets, showing that RNO remains both accurate and computationally efficient.
>
>
> Table1: Results on Reaction-Diff. benchmarks.
> | Model        | Params (M)  | Train time (s/epoch)  | Peak memory  | Relative ℓ₂  |
> |--------------|--------------|-------------------------|----------------|----------------|
> | DeepONet     | 567,809      | 0.263                   | 246.30 MB      | 0.7969         |
> | FNO          | 310,913      | 0.098                   | 230.67 MB      | 0.1214         |
> | WNO          | 192,833      | 0.112                   | 281.11 MB      | 0.1713         |
> | LNO          | 64,241       | 0.298                   | 1,635.57 MB    | 0.1355         |
> | RNO (ours) | 172,229 | 0.052               | 111.17 MB  | 0.0899     |
>
> Table2: Results on Diffusion benchmarks.
> | Model        | Params (M)  | Train time (s/epoch)  | Peak memory  | Relative ℓ₂  |
> |--------------|--------------|-------------------------|----------------|----------------|
> | DeepONet     |  403073     | 0.285                   |  365.67 MB      | 0.1356         |
> | FNO          | 527,745      | 0.278                   | 313.36 MB      | 0.0229         |
> | WNO          | 192,833      | 0.112                   | 281.11 MB      | 0.0179         |
> | LNO          | 76,369       | 0.512                   | 8,805.78 MB    | 0.0081         |
> | RNO (ours) | 150,609 | 0.462               | 245.71 MB  | 0.0079
>
>
> [1] Quaternionic representation of the riesz pyramid for video magnification, DSpace@MIT2014.
>
> [2] Riesz pyramids for fast phase-based video magnification, ICCP2014.

---

> ### Author Response · Authors · 2025-11-24
>
> ---
> >**Q2**:
> > Can you differentiate your method from FNO + CNN?.
>
> **A2**:
> Thank you for raising this question; it is very helpful for clarifying the distinctions between RNO and an FNO+CNN design. Our response is summarized below.
>
> * 2.1. **Applied perspective: resolution generalization.**
>    A key advantage of FNO is that it learns an operator on a **continuous function** space and can be evaluated at different resolutions; the Fourier modes, rather than fixed discrete matrices, govern its behavior. Once a CNN branch is attached, this property is generally lost, since the CNN is tied to a specific grid resolution and boundary handling. RNO, by contrast, **preserves the integral operator structure of FNO and adds directional derivatives only through spectral multipliers and pointwise mixing**, so it inherits the same resolution flexibility. RNO is conceptually closer to LNO: it extracts richer and more orthogonal information from the signal itself, focusing on directionality intrinsic to the input rather than architectural concatenation. This design also provides a degree of physical interpretability.
>
> * 2.2. **Structural perspective: principled locality vs. architectural concatenation.**
>    FNO+CNN introduces locality via a generic convolutional branch, **without** controlling what frequencies or orientations are emphasized. RNO instead uses Riesz-based directional derivatives, which are rotation-covariant and directly related to the phase gradient ∇φ (A.1). The derivative channels are thus **non-redundant** and physically meaningful, providing a structured way to enrich the high-frequency spectrum without sacrificing operator properties. We formalize this via a Green’s function[2] view of RNO in A.4. Our goal is precisely to enhance operator expressiveness *without* importing heavy external structures or compromising core operator properties.
>
> Empirically, we also compare against two strong FNO+localConv variants, E-FNO [2] and loglo-FNO [3] (Sec. 4.2, A.8). As summarized in Tables, both are slower and less accurate than RNO on Reaction–Diff. and Brusselator benchmarks, while using 4–5× more parameters.
>
> Table3: Model efficiency on Reaction-Diff. dataset.
> | Model        | Params  | Train time (s/epoch)  | Peak memory (MB)  | Relative ℓ₂  |
> |--------------|---------|-------------------------|---------------------|----------------|
> | E-FNO        | 793,985 | 0.166                   | 241.07              | 0.1131         |
> | loglo-FNO    | 764,225 | 0.142                   | 617.22              | 0.1009         |
> | RNO (ours) | 172,229 | 0.052             | 111.17          | 0.0899     |
>
>
> Table4: Relative ℓ₂ results on Reaction-Diff. and Brusselator dataset.
> | Model      | Reac–Diff. | Brusselator |
> |------------|------------|-------------|
> | WNO        | 0.1713     | 0.1789      |
> | LNO        | 0.1355     | 0.1858      |
> | loglo-FNO  | 0.1009     | 0.1679      |
> | E-FNO    | 0.1131     | 0.1703      |
> | RNO (ours) | 0.0899 | 0.1317 |
>
> [1] Neural operator: Graph kernel network for partial differential equations. 2020arXiv.
>
> [2] Enhancing Fourier Neural Operators with Local Spatial Features, arxiv2025.
>
> [3] LOGLO-FNO: Efficient Learning of Local and Global Features in Fourier Neural Operators, arxiv2025.

---

> ### Author Response · Authors · 2025-11-24
>
> ---
> >**Q3**:
> > Motivation of derivatives. The authors motivate their architecture by emphasizing the importance of derivatives in solving PDEs. While this is a generally safe statement, the characteristics of different PDE classes makes it less convincing. There are many (spatially) localized PDEs, but there are also PDEs that are global, or even scale-local. Motivating an architecture based on a vague view of all PDEs does not convince me. Nevertheless, I acknowledge that spectral bias is a very significant deficiency of neural operators. I suggest the authors to emphasize on how RNO properly models the full spectrum, why different rank-reduction method are useful for global vs local dynamics.
>
>
> **A3**:
> Thank you for raising this point.
> * 3.1 We acknowledged that the original motivation was overly broad, as different PDEs exhibit different characteristics. We have therefore added clarification in the 'Problem Definition' section. The core idea of our method is that **local dynamics** — captured through derivatives — complement and refine global representations. Derivatives are fundamental components of many PDEs, that contains rich small-scale variation, and this information can be incorporated in a manner analogous to a Taylor expansion. To strengthen this connection, we added a discussion linking Taylor expansion, optical flow, and PDE structure in the A.9.
>
> * 3.2 From a signal-processing perspective, our method emphasizes how derivative information enriches global features by providing fine-scale detail, enabling the model to better extract higher-order variations that **naturally exist** in the data. To support this observation, we also added a qualitative comparison of RNO on higher-order quantities in the Navier–Stokes equations at Re = 5000, such as vorticity (∇×u) and divergence (∇·u), which is shown in Fig. 10. RNO shows markedly better performance in predicting higher-order quantities of the NS equations.
>
> We hope this clarifies your concerns. We are committed to thoroughly incorporating your suggestions in the next version of the paper. Thank you once again for your valuable feedback.

---

### Author Response · Authors · 2025-11-24

**General Response**

We deeply appreciate the insightful and valuable comments provided by all reviewers.

We are grateful for the reviewers’ recognition that this paper proposes a **physically grounded** neural operator architecture that blends global with **Riesz-based directional derivatives** to better capture **local non-stationarity** in PDEs, and that it provides a comprehensive and carefully controlled evaluation across diverse synthetic and real-world benchmarks.

Overall, we are encouraged by the positive assessments, which emphasize that:

* RNO introduces a novel way to combine global modes with speactral directional derivatives, supported by a clear physical and theoretical motivation.(highlighted by Reviewers F9gf, Pgop, xBjE)
* The paper offers a broad and carefully designed evaluation across multiple PDE benchmarks and a realistic ERA5 task, showing strong performance and generalization.(by Reviewers UKYE, F9gf, xBjE)
* The theoretical foundations of the paper are solid, and the exposition is clear.(by Reviewers F9gf, xBjE）

To address the main concerns, we have added several experiments and clarifications:

* We have further strengthened the theoretical analysis to to further reinforce the advantages and validity of our approach in A.2, A.3, A.4, A.8, A.9.
* More comprehensive experiments, including efficiency analysis, stronger baseline comparisons, additional visualizations, higher-order evaluations, robustness tests, and more in A.12, A.13, A.14, A.15.
* We have added more detailed explanations of the finer aspects of our method, including pseudocode，derivative mechanism and benchmark details in A.16, A17, A18.7.

All revisions are marked in blue. Below, we respond to each reviewer’s comments in detail.

---

### Author Response · Authors · 2025-11-30
**Summary of Revisions for the Area Chair**

Dear Area Chair,

Thank you for your dedicated service to the community. We briefly summarize our main revisions and point to where changes were made.

Our work strengthens classical neural operators without adding heavy external architectures (Transformers, CNNs etc) that can break **resolution invariance** and **physical continuity**. Instead, we enrich operators with **Riesz-based directional derivatives**, improving their ability to handle non-stationary and complex physical data.

Reviewers highlighted three main strengths:
* (i) a novel mechanism to couple global modes with spectral directional derivatives, with clear physical/theoretical motivation (Reviewers_F9gf, Pgop, xBjE);
* (ii) a broad evaluation across multiple PDE benchmarks and ERA5, with strong performance and generalization (Reviewers_UKYE, F9gf, xBjE);
* (iii) solid theoretical foundations and generally clear exposition (Reviewers_F9gf, xBjE).

To address the main concerns, we expanded the paper in three directions: **theory** (stronger analysis in A.2, A.3, A.4, A.8, A.9), **experiments** (efficiency analysis, stronger baselines, extra visualizations, higher-order evaluations, robustness tests in A.12–A.15), and **implementation** (code in supplementary material, pseudocode, derivative mechanisms, and benchmark details in A.16–A.18). We include a table of the appendix to help you locate the content.

The following is to help you quickly navigate our responses.

---
>**Reviewer UKYE**

- **Complexity of the Riesz neural operator.**
  We added a **comprehensive efficiency analysis** for RNO, showing that it remains highly efficient. Meanwhile, we added a complexity and efficiency analysis for spatial FIR vs. FFT-based variants (O(Nk²) vs. O(N log N)), plus pseudocode (Sec. 4.2, Fig. 6, App. A.16).

- **Relation to FNO+CNN.**
  We clarified that RNO preserves NO’s **continuous-operator / resolution-generalization** properties by injecting locality only via spectral Riesz multipliers, and we added FNO+localConv baselines (E-FNO, loglo-FNO) showing RNO is both more accurate and more efficient (Sec. 3, Sec. 4.2, Apps. A.4, A.8, A.16).

- **Motivation around derivatives.**
  We narrowed the motivation to PDEs where local dynamics refine global modes, formalized the **first-order Taylor view of derivative channels**, and supported this with **Navier–Stokes vorticity/divergence experiments** (problem definition, App. A.9, Fig. 10).

---
>**Reviewer F9gf**

- **Higher-order behavior.**
  We clarified that RNO implements a controlled **first-order Taylor approximation**, derived a wavelength-dependent error bound, tuned derivative scales (Apps. A.2, A.6), and added ∇u, Δu, vorticity, and divergence evaluations on Navier–Stokes where RNO outperforms baselines (App. A.13, Fig. 10).

- **Efficiency.**
  See Q1 of Reviewer UKYE (Sec. 4.2, App. A.16).

- **Stability and noise robustness.**
  We emphasized the L²-bounded nature of the Riesz transform, introduced a bounded scaling factor ζ, and added **noise-robustness experiments** where RNO remains stable and competitive or better than baselines (Apps. A.1, A.6, A.15, plus ERA5).

---
>**Reviewer Pgop**

- **Baseline fairness and discrepancies.**
  To address discrepancies (e.g., LNO on Beam/Duffing), we re-ran all baselines under a **unified protocol** and added **loss curves** and **hyperparameter details** (App. A.16, Fig. 11).

- **Navier–Stokes rigor.**
  We strengthened the NS experiments by adding WNO and F-FNO and showed RNO achieves the best errors at Re = 40, 500, and 5000, with large gains at high Re (Sec. 4 NS table, Fig. 10).

- **Theory, high-frequency baselines, and code.**
  We tightened the theoretical claims (Intro, App. A.2), added **high-frequency baselines** (LNO, WNO, E-FNO, loglo-FNO) where RNO attains the lowest relative ℓ₂ on Reaction–Diff. and Brusselator (Sec. 4.2, App. A.8), and provided complete **code** of NS data in the supplementary material.

---
>**Reviewer xBjE**

- **WNO comparisons**
  We added WNO as a core baseline across every experiments, showing RNO is competitive or superior (Sec. 4, high-frequency + ERA5 tables).

- **Algorithmic details and ERA5 setup**
  We provided full **pseudocode** for spatial and spectral RNO variants (Riesz computation, monogenic features, directional mixing, training loop) in App. A.16, and made ERA5 self-contained with data sizes, splits, preprocessing, metrics, and visualizations (App. A.18).

- **Learning directional derivatives**
  We clarified how directional derivatives are learned via **multi-orientation** frequency components and **adaptive local-to-global fusion** (method subsection, App. A.9).

Although the incident prevented further reviewer discussion, we have provided careful point-by-point responses to all concerns, addressing doubts and supporting all aspects of the work.

We extend our heartfelt thanks to the Area Chair and all reviewers for their efforts and valuable feedback. Thank you!

---

### Meta-Review · Area_Chair_14fb · 2026-01-09

**Summary:**

Reviewers indicate that:
1. Using the Reisz transform to extract information about directional derivatives is novel.
2. Space-time complexity of the method is not reported on the numerical experiments and is not analyzed theoretically.
3. Some important baselines are missing from the numerical experiments.
4. The method is conceptually similar to using an FNO with CNN to help capture local features.
5. The numerical experiments are extensive, covering a wide variety of physical problems.

**Reviewer Concerns:**

The authors have addressed the majority of reviewer concerns. They have added extra baselines, new numerical experiments, showed empirical results for the method's space-time complexity, added extra theory, and clarified the novelty of the method. It seems to me that essentially all reviewers issues are addressed.

**Reviewer Scores:**

Given the extensive rebuttal by the authors, reviewers Pgop and UKYE would increase their scores.

---

### Decision · Program_Chairs · 2026-01-26

Accept (Poster)